# Antimicrobial Photodynamic Therapy: Latest Developments with a Focus on Combinatory Strategies

**DOI:** 10.3390/pharmaceutics13121995

**Published:** 2021-11-24

**Authors:** Raphaëlle Youf, Max Müller, Ali Balasini, Franck Thétiot, Mareike Müller, Alizé Hascoët, Ulrich Jonas, Holger Schönherr, Gilles Lemercier, Tristan Montier, Tony Le Gall

**Affiliations:** 1Univ Brest, INSERM, EFS, UMR 1078, GGB-GTCA, F-29200 Brest, France; raphaelle.youf@univ-brest.fr (R.Y.); alz.hascoet@gmail.com (A.H.); tristan.montier@univ-brest.fr (T.M.); 2Physical Chemistry I & Research Center of Micro- and Nanochemistry and (Bio)Technology of Micro and Nanochemistry and Engineering (Cμ), Department of Chemistry and Biology, University of Siegen, Adolf-Reichwein-Straße 2, 57076 Siegen, Germany; max.mueller@uni-siegen.de (M.M.); m.mueller@chemie-bio.uni-siegen.de (M.M.); 3Macromolecular Chemistry, Department of Chemistry and Biology, University of Siegen, Adolf-Reichwein-Straße 2, 57076 Siegen, Germany; alibalasini94@gmail.com (A.B.); jonas@chemie.uni-siegen.de (U.J.); 4Unité Mixte de Recherche (UMR), Centre National de la Recherche Scientifique (CNRS) 6521, Université de Brest (UBO), CS 93837, 29238 Brest, France; 5Coordination Chemistry Team, Unité Mixte de Recherche (UMR), Centre National de la Recherche Scientifique (CNRS) 7312, Institut de Chimie Moléculaire de Reims (ICMR), Université de Reims Champagne-Ardenne, BP 1039, CEDEX 2, 51687 Reims, France; 6CHRU de Brest, Service de Génétique Médicale et de Biologie de la Reproduction, Centre de Référence des Maladies Rares Maladies Neuromusculaires, 29200 Brest, France

**Keywords:** antimicrobials, ROS, combinatory strategies, photodynamic therapy, multidrug resistance, nanoparticles, photosensitizers

## Abstract

Antimicrobial photodynamic therapy (aPDT) has become a fundamental tool in modern therapeutics, notably due to the expanding versatility of photosensitizers (PSs) and the numerous possibilities to combine aPDT with other antimicrobial treatments to combat localized infections. After revisiting the basic principles of aPDT, this review first highlights the current state of the art of curative or preventive aPDT applications with relevant clinical trials. In addition, the most recent developments in photochemistry and photophysics as well as advanced carrier systems in the context of aPDT are provided, with a focus on the latest generations of efficient and versatile PSs and the progress towards hybrid-multicomponent systems. In particular, deeper insight into combinatory aPDT approaches is afforded, involving non-radiative or other light-based modalities. Selected aPDT perspectives are outlined, pointing out new strategies to target and treat microorganisms. Finally, the review works out the evolution of the conceptually simple PDT methodology towards a much more sophisticated, integrated, and innovative technology as an important element of potent antimicrobial strategies.

## 1. Introduction

Antimicrobial resistance (AMR) occurring in bacteria, viruses, fungi, and parasites is a global health and development threat, declared by the WHO as one of the top 10 global public health threats facing humanity. The misuse and overuse of antimicrobials make almost all disease-causing microbes resistant to drugs commonly used to treat them [1]. Multidrug resistance (MDR) to critical classes of antibiotics has gradually increased in nosocomial pathogens, including *Enterococcus faecium*, *Staphylococcus aureus*, *Klebsiella pneumoniae*, *Acinetobacter baumannii*, *Pseudomonas aeruginosa*, and *Enterobacter* spp. (which are gathered in the so-called ESKAPE group) [2]. Currently, in Europe, AMR is estimated to be responsible for 33,000 deaths every year and the annual economic toll, covering extra healthcare costs and productivity losses, amounts to at least EUR 1.5 billion [3,4]. Unless adequately tackled, 10 million people a year will die from drug-resistant infections by 2050, according to the predictions of the government-commissioned O’Neill report [5].

With the decline in the discovery of new antimicrobials since 1970s, the mainstream approach for the development of new drugs to combat emerging and re-emerging resistant pathogens has focused on the modification of existing compounds. However, one key recommendation encourages stimulation of early stage drug discovery [6]. Among emerging antimicrobial therapeutic alternatives, light-based approaches show particular promise [7]. Traditionally, phototherapy was already a common practice in ancient Greece, Egypt, and India to treat skin diseases [8]. At the beginning of the 20th century, Oscar Raab first described the phototoxicity of the dye acridine red against *Paramecium caudatum,* and Tappenier and Jesionek reported the photodynamic effects of eosin suitable for treating diverse cutaneous diseases. Since then, PDT was established as the administration of a non-toxic photosensitizer (PS) followed by exposure to light irradiation at an appropriate wavelength focused on an area to treat [7]. While anti-cancer PDT is a clinical reality for 25 years [9], PDT as an antimicrobial treatment was demonstrated for the first time against drug-resistant infections in the healthcare sector in the early 1990s, leading to a “photo-antimicrobial renaissance era” [7]. Major MDR bacteria have been found susceptible to antimicrobial PDT (aPDT), independently of their drug-resistance profiles [10,11]. To date, resistance to aPDT is rarely reported, indicating that the possibility for microbes to adapt and escape this treatment can occur but is still contained. More effective aPDT systems are continuously developed, notably via combinatory approaches using multiple chemical systems and/or modalities. At the current stage of development, aPDT cannot address systemic infections but it holds great promise for treating localized infections and to fight AMR.

While excellent earlier authoritative reviews provide a detailed description of aPDT [12,13,14,15], the present review focuses on most recent developments in the field for the last 5 years, with a focus on aPDT combinatory strategies. It should be noticed that aPDT is sometimes referred to as photodynamic antimicrobial chemotherapy, light-based antimicrobial therapy, photo-controlled antimicrobial therapy, or antimicrobial photo-inactivation. In this review, all these synonyms were considered to present (i) the current state of aPDT applications in preclinical and clinical settings, (ii) a state of the art with recent developments in photosystems, (iii) the implementation of multicomponent nanotechnologies and recent molecular engineering, and (iv) the exploration of combinatory aPDT approaches towards possible future therapeutic innovations.

## 2. PDT: General Presentation and Features

### 2.1. Photochemical Pathways and Reactive Oxygen Species Production

Generally speaking, a given PS has the potential to produce reactive oxygen species (ROS) under specific conditions (Figure 1A). Typically, it possesses a stable electronic configuration called ground state level. Following irradiation and absorption of a photon, the PS is converted from a low (fundamental) energy level (^1^PS) to a “Frank Condon” short-lived, very reactive, excited singlet state ^1^PS* [16,17,18]. Subsequently, the PS can lose energy by emitting fluorescence or heat via internal conversion (IC), thereby returning to its initial ground state level; alternatively, it can be converted by a so-called inter-system crossing (ISC) to a longer-living excited triplet state ^3^PS*. From this state, two types of chemical reaction pathways can occur, known as Type I electron transfer and Type II energy transfer, which can take place simultaneously [19]. In the Type I reaction, the ^3^PS* captures an electron (e^−^) from a reducing molecule (R) in its vicinity, which induces an electron transfer producing the superoxide anion radical (O_2_^●−^) and, after the subsequent reduction, leads to the generation of more cytotoxic ROS including hydrogen peroxide (H_2_O_2_) and hydroxyl radical (HO^●^). In the Type II reaction, a direct energy transfer occurs from the ^3^PS* to the ground state molecular oxygen (^3^O_2_) that is then converted to singlet oxygen (^1^O_2_). The ROS thus produced encompass O_2_^●^^−^, H_2_O_2_, HO^●^, and ^1^O_2_, the last two being the most reactive and most cytotoxic species but also those with the shortest diffusion distance. One PS molecule can generate thousands of molecules of ^1^O_2_, depending notably on its ^1^O_2_ quantum yield, the surrounding environment, and the respective occurrence of Type I and Type II mechanisms [12,17,20].

In addition to the above-mentioned Type I and Type II mechanisms, Hamblin et al. recently proposed introduction of a Type III photochemical pathway, following which the radical anion PS^•−^ and/or inorganic radicals formed in absence of oxygen could also lead to photoinactivation [21]. These authors indeed identified several circumstances in which oxygen-independent photoinactivation of bacteria using specific PSs can be obtained.

### 2.2. Biological Effects of aPDT: Potential Targets and Related Mechanisms

The main first targets of aPDT are external microbial structures, i.e., the cell wall, cell membrane, or virus capsid and envelope [22,23]. Photodynamic inactivation (PDI) can be obtained against microorganisms growing as planktonic cells and/or in biofilms [13]. In biofilm matrices, the diffusion of PSs can be delayed or PSs can be sequestered, in spite of photodamage induced on various components such as polysaccharides and extracellular DNA [24,25]. The diffusion potential of ROS depends on (i) the maximal time-limited diffusion length, especially for ^1^O_2_ that possesses a shorter half-life compared with other ROS, (ii) the photostability in a given environmental medium, and (iii) the chemical properties of PSs (e.g., molecular size, charge, lipophilicity, stability), which influence the interactions of the latter with target microorganisms [26]. Photoinactivation of Gram(+) bacteria can be obtained with a given PS, irrespective of its charge, whereas that of Gram(−) bacteria generally requires a cationic PS, or a combination of a neutral PS with membrane-disrupting agents [27].

Internalization of PSs in prokaryotic or eukaryotic cells can also occur, thus causing various intracellular oxidative damage (such as in organelles in eukaryotic cells, e.g., nucleus and mitochondria in fungal cells) [28]. To protect their intracellular components, microbial cells can induce the production of antioxidant defenses such as protective enzymes (such as superoxide dismutase (SOD), catalase and glutathion (GSH)-peroxidase) or pigments (such as carotenoids acting as nonphotochemical ^1^O_2_ quenchers). Nevertheless, these mechanisms can be insufficient to thwart aPDT-induced oxidative stress because intracellular components (including antioxidant defenses) can be also irreversibly photodamaged by ROS [29]. The latter can act on the DNA level through two mechanisms, i.e., alteration or modulation. Breaks in single-strand and double-strand DNA, and the disappearance of the super-coiled form of plasmid DNA have been reported in both Gram(+) and Gram(−) species. Indeed, PSs can interact with nucleic acids via electrostatic interactions and induce reduction of guanine residues causing DNA cleavage [30]. Again, microorganisms can induce the overproduction of proteins involved in the repair of photodamaged DNA; however, some bacteria, such as *Helicobacter pylori*, possess only a few of such protective repair mechanisms [31]. Upon PDT treatment, ROS and ^1^O_2_ can modify bacterial gene expression profiles by modulating (i) the quorum sensing pathway, therefore inhibiting biofilm formation as shown in in vitro models, and/or (ii) the anti-virulence activities by reducing the gene expression of virulence factors in diverse clinical pathogens [32,33,34,35].

Given the multi-targeted nature of aPDT, the possibility for microorganisms to develop resistance is supposed to be very limited [36]. However, they may be able to respond to aPDT in different ways. For instance, light response adaptation can occur in some microorganisms, such as *E. coli* upon exposure to blue light inducing the production of a biofilm polysaccharide colonic acid [37]. Moreover, some PSs are substrates of efflux systems that may be overproduced; specific inhibitors of the latter can be used to restore phototoxicity [38,39,40]. After sublethal aPDT, the biofilm-forming ability of bacteria can increase, making them less susceptible to the same treatment [41]. Each strategy should thus be carefully examined with regard to the ability of target microorganisms to adapt and escape treatments. The latter may be noticeably reduced when using at the same time multiple antimicrobial molecular partners and modalities (read below).

### 2.3. Important Parameters and Requirements for an “Ideal” aPDT

According to Cieplik et al. [12], an “ideal” aPDT system should meet the following set of general requirements: (i) PS physicochemical properties: efficient PSs for aPDT possess most frequently a high hydrophilicity index and at least one cationic charge promoting interactions with pathogens, especially Gram(−) bacteria; (ii) PS photosensitivity: following irradiation, a good PS produces a high rate of cytotoxic oxygen species (Figure 1A), depending notably on its ^1^O_2_ quantum yield, its stability, and the environmental media; (iii) light source parameters: efficient irradiation of PSs must take into account a coherent light exposure ensuring a good transmittance efficiency with no side effects; (iv) safety: the PS has to be specific to target microorganism(s), inducing insignificant or only a few side effects for the host, including no or few immunity responses; and (v) ease for implementation in clinical practice: aPDT has to be relatively easy to use (due to the rapid, non-aggressive, and non-invasive light application), cost-effective, and accessible. It is obvious that the detailed specification of requirements can vary depending on the target applications. The improvement of PSs is an ongoing challenge that implies moving towards more rational chemical engineering and biological investigations [11,42], as discussed in this review.

## 3. Positioning of aPDT in Current Human Healthcare Treatments

Over the past years, the number of studies dealing with aPDT has dramatically increased, emphasizing the potential of this therapeutic approach to treat a broad spectrum of microorganisms including bacteria, fungi, viruses, and parasites (Figure 1B). In this part, recent in vitro screening studies, preclinical (using animal models) and clinical investigations are briefly reviewed. Details about the PSs involved and their structures are provided in Section 4 “State of the art with recent photo(nano)system developments”.

### 3.1. Curative Preclinical aPDT

#### 3.1.1. Treatment of Bacterial Infections

PDT is a promising alternative approach to antibiotherapy for photoinactivating a broad spectrum of bacterial pathogens, either Gram(+) or Gram(−), responsible for diverse infections in humans. The antibacterial versatility of PDT can be highlighted in different ways, notably by considering lists of critically important human pathogens. First, in recent years, more attention has been paid to the potential of aPDT to fight against bacteria involved in hard-to-treat infections, especially those forming the ESKAPE group [2,43]. Second, other critical pathogen lists can be considered, such as the NIAID emerging infectious diseases/pathogens category that includes biodefense research and additional emerging infectious diseases/pathogens. To our knowledge, the susceptibility of more than 50% of the bacteria in the NIAID critical pathogen list has been considered in at least one aPDT study. In other words, bacteria causing the worst endemic infections including anthrax, botulism, melioidosis, cholerae, plague, and tuberculosis have already been considered. On the opposite, vector-borne diseases transmitted by human parasites, such as *Borrelia mayonii* and *Bartonella henselae* have not been addressed in that regard yet. Multiple experimental settings have been considered to demonstrate the potential of aPDT to photoinactivate pathogenic bacteria, growing as planktonic forms, but also in biofilm matrices, and using diverse animal models [44,45]. Among human pathogens, bacteria implicated in oral infections, especially cariogenic, periodontic, and endodontic injuries, have probably been the most intensely investigated [46]. Although less considered, other indications have also been evaluated with aPDT, including osteomyelitis, meningitis, pneumonia, lung abscess, and emphysema [47,48].

#### 3.1.2. Treatment of Fungal Infections

Fungal infections caused by invasive candidiasis are widely recognized as a major cause of morbidity and mortality in the health care environment [49]. In addition to the opportunistic features of some fungi, resistance to first-line antifungals (such as echinocandins and fluconazole) is spreading, compromising the efficiency of conventional antifungal therapies. Despite the fact that yeasts are naturally more resistant to PDT than bacteria, noticeable in vitro and in vivo effects against fungal infections have been reported, including germ load reduction, biofilm inhibition and clearance, and eradication of persistent colonization [50,51,52,53]. Furthermore, antifungal PDT has demonstrated its potential as an adjunctive (potentially synergistic) treatment procedure to the conventional fungicide Nystatin [54]. Chen et al. identified and summarized other fungal infections that may be treated with aPDT, including *onychomycosis*, *tinea cruris*, *pityriasis versicolor*, *chromoblastomycosis*, and the cutaneous *sporothricosis* [55]. Recently, other fungal infections, such as fungi associated to *mucormycosis* (recognized as emerging critical pathogens in the NIAID) were successfully treated with PDT [56].

#### 3.1.3. Treatment of Viral Infections

Although vaccines have drastically reduced the spreading of some of the most virulent viruses around the world, antiviral research and development remain a healthcare priority notably due to emerging viral infectious diseases [57]. PDT holds promises to help treat the latter, as well as other viruses implicated in complications of some cancers. The oldest, but also the most current, application of antiviral PDT concerns the decontamination of blood products potentially containing hepatitis B/C or West Nile virus [23,58]. The PDI of viral infections was explored in many studies considering other various viruses including arbovirus, SV40, poliovirus, encephalitis virus, phages, and HSV [59,60]. In addition, emerging viruses such as Zika, Ebola, or Tickborne hemorrhagic fever viruses have been considered [23], as well as viruses responsible for epidemic/pandemic crises such as influenza virus, SARS-CoV-2 virus, and its mutants/variants [61,62].

#### 3.1.4. Treatment of Parasite Infections

Drug resistance is also rapidly spreading in parasites. For example, resistance to artemisinin (which is used to treat plasmodium infections causing malaria) increases drastically, even when combined with other drugs (WHO, https://www.who.int/news-room/fact-sheets/detail/antimicrobial-resistance, accessed date: 1 September 2021). Antiparasital effect of PDT was demonstrated toward critical parasites in public health such as tropical pathogens including Leishmania, African trypanosoma, and Plasmodium [63,64]. Another way to limit the propagation of the vectors is to rely on photoinductible biolarvicides. Such an approach was investigated in order to control Aedes, Anopheles, and Culex, which are tropical disease-carrying species [65,66]. This was also investigated in Lyme disease using safranin-PDT for reducing the reproduction of ticks [67].

#### 3.1.5. Treatment of Polymicrobial Infections

Quite recently, interest has grown to explore the potential of aPDT against polymicrobial infections involving multispecies pathogens. A study demonstrated that the susceptibility to PDI of *S. aureus* and *C. albicans* growing in mixed biofilms is lower compared with single-species biofilms, which may be due to the difference in the chemical composition and viscosity of the composed matrix [24]. Nevertheless, aPDT applications are of interest regarding hard-to-treat infections due to polymicrobial biofilms colonization, such as chronic rhinosinusitis. They could effectively be eradicated by aPDT in a maxillary sinus cavity model [68]. Moreover, Biel et al. showed that aPDT can eradicate polymicrobial biofilms in the endotracheal tubes, which are factors leading to ventilator-associated pneumonia [69]. More recently, aPDT was shown capable of significantly improving wound healing in mice with polymicrobial infections [70]. However, such an approach remains a challenge due to the respective affinity of PSs to each species being usually reduced in polymicrobial systems.

### 3.2. Current Clinical aPDT Practices

Many clinical trials have been done for evaluating aPDT approaches in the treatment of bacterial/fungal oral infections. This is facilitated by the development of easy to use light sources in dentistry. On the opposite, it can be compromised by the development of persistent (multispecies) biofilms. Skin infections such as *Acnes vulgaris,* caused by *Propionibacterium acnes*, was one of the first microbial infections to reach the stage of aPDT clinical trial. A few clinical trials demonstrated that onychomycosis, such as *tinea cruris*, *tinea pedis*, and interdigital mycoses, could be treated with aPDT. Results demonstrated that aPDT is effective and well-tolerated, but infections can recur frequently [71,72]. Among cutaneous infections, non-healing chronic wounds in patients with chronic leg and/or foot ulcers were efficiently treated with aPDT, inducing a significant reduction in microbial load (even immediately after the treatment), a better wound healing, and no safety issues [73,74]. Osteomyelitis in patients with chronic ulcers can be treated with aPDT to prevent gangrene and amputations in the extremities of diabetic patients [75]. Clinical studies for treating *H. pylori* in gastric ulcers can be conducted using phototherapy without any PS administration; *H. pylori* naturally accumulates intracellular PSs (porphyrins) and therefore could be inactivated by phototherapy thanks to an ingenious blue/violet light delivery system [76]. A list of recently closed aPDT clinical trials is provided in Table 1; many other trials (not shown here) are still ongoing.

### 3.3. Toward Preventive/Prophylactic Treatments

Beside curative antimicrobial treatments, PDT may be also used to decontaminate medical equipment and tools in hospitals, for preventive/prophylactic aims [27]. For example, photoantimicrobial textiles were reported to efficiently photoinactivate bacteria and viruses, suggesting that self-sterilizing medical gowns could be developed [77]. PDT can also decontaminate medical tools similarly to conventional chemical agents, as demonstrated in a recent comparative study [78]. Its application for the decontamination of routine informatics tools, office equipment, or packing materials demonstrates sterilization potential that could be useful to avoid hospital-acquired infections and to protect healthcare workers. Furthermore, photodisinfection of water and photoinactivation of food-borne pathogens can bring substantial benefits to people’s daily lives [65].

## 4. State of the Art with Recent Photo(nano)System Developments

### 4.1. Single PSs

#### 4.1.1. Organic PSs and Their Derivatives

Organic PSs used in aPDT have been well described in some recent reviews [79,80,81] (Figure 2A). Briefly, since the first use of eosin in 1904, various PSs were investigated, especially in the phenothiazinium group, which includes methylene blue (MB) and toluidine blue O (TBO). Thanks to an absorption spectrum in the red region of light, these PSs can be effective in tissues while being less toxic than other PSs. Their aPDT properties are mostly due to a high ROS production following Type I mechanism (Figure 1A). Structural derivatives have been also reported, including new MB and dimethyl-MB [42]. Another group gathers compounds featuring a macrocyclic structure composed of pyrroles, such as porphyrins and its precursor 5-aminolevulinic acid (5-ALA), phthalocyanines, and chlorins. Macrocyclic compounds are generally hydrophilic, positively charged, and they exhibit a good singlet oxygen quantum yield. Modifications of their chemical structure were intensively studied, especially for porphyrins and phthalocyanines [82,83,84,85,86,87]. The halogenated xanthenes gather PSs with a structure similar to that of fluorescein. Among them, eosin Y, erythrosine, and rose bengal (RB) were the most studied [88,89,90]. These compounds are anionic, which can limit their interaction with bacterial cells and their aPDT effect in spite of good singlet oxygen quantum yields. Natural compounds, including coumarins, furanocoumarins, benzofurans, anthraquinones, and flavin derivatives are often found in plants and other organisms. They are characterized by an absorption spectrum in white light or UVA. The most used are curcumin, riboflavin, and hypericin [7,80]. Nanostructures such as fullerenes are interesting PSs because of their ability to modulate Type I and Type II reactions (Figure 1A), depending on the near environment and the light source applied [91,92]. In this family, some quantum dots (QDs) can act as photoantimicrobials [80]. Other synthetic fluorescent dyes such as organoboron compounds (e.g., boron–dipyrromethene (BODIPY)), and cyanine dyes (e.g., indocyanine green (ICG)) are known for their high photostability, high extinction coefficients, and high fluorescence quantum yields [93].

Over the past five years, some new organic PSs were reported. Among them, optimized natural PSs such as anthraquinones and diacethylcurcumin can be listed. Others include derivatives of synthetic dyes such as monobrominated neutral red or azure B [79,80,81] (Figure 2A).

#### 4.1.2. Coordination and Organometallic Complexes-Based PSs

Distinctly from metal nanoparticles (NPs; see Section 4.2.1 “Metal-based systems”), metal complexes, either coordination or organometallic complexes, are of increasing interest as PSs in PDT [95] (Figure 2B). They generally consist of a central metal core combined with ligands, involving coordinate covalent bonds (in coordination compounds) or at least one metal–carbon bond (in organometallic compounds). Compared with organic compounds, metal complexes have been notably much less considered and still remain largely underexploited regarding their potential use as new antibiotics [96]. Frei et al. recently reported on the antimicrobial activity of 906 metal-containing compounds. They showed that, considering antimicrobial activity against critical antibiotic-resistant pathogens, metal-bearing compounds displayed hit rates about 10 times higher than purely organic molecules [97].

Metal complexes display a panel of specific properties that make them promising PSs candidates. The variety of metal ions and ligands can be assembled in scaffolds featuring very diverse geometries [97,98]. Whereas most organic PSs are linear or planar molecules, metal complexes can exhibit much more complex—three-dimensional—geometries, which can improve interaction and molecular recognition with cellular targets, enlarge the spectrum of activity, and impact on biological fate [98,99,100]. Furthermore, the modulation of the design of metal complexes allows to fine tune their hydrophilic–lipophilic balance, solubility, photophysical properties, and eventual “dark toxicity” (i.e., the toxicity in the absence of specific irradiation). Metal complexes can display many excited-state electronic configurations associated with the central metal, the ligands, or involving both the metal and the ligand(s) in charge-transfer states (metal-to-ligand charge transfer or ligand-to-metal charge transfer). Although it is not always considered, the investigation of excited states of metal complexes (Figure 1A) is however of prime importance. Triplet states can be more easily accessed due to the enhanced spin-orbit coupling induced by the presence of the heavy metallic atom. Compared with natural PSs, metal complexes can act, besides ROS generation, via other mechanisms including redox activation, ligand exchange, and depletion of substrates involved in vital cellular processes [96,97,101].

Besides their first intended development as anticancer compounds, metal complexes have also been envisaged as potential “metallo-antibiotics”, benefiting from the knowledge accumulated about their chemical properties and biological behaviors [102]. Quite recently, several metal compounds were characterized for their activity in aPDT. For instance, platinum(II), molybdenum(II), ruthenium(II), cobalt(II), and iridium(III) were proposed as new classes of stable photo-activatable metal complexes capable of combating AMR [11,103,104,105,106,107,108]. In particular, many mononuclear and polynuclear Ru(II/III) complexes have been considered as potential antibiotics, antifungals, antiparasitics, or antivirals, which have been recently extensively reviewed [107]. It is worth noticing here that, within a series of 906 metal compounds, ruthenium was the most frequent element in active antimicrobial compounds that are nontoxic to eukaryotic cells, followed by silver, palladium, and iridium [97]. Ru(II) polypyridyl complexes were assayed in several aPDT studies. For instance, Ru(DIP)_2_(bdt) and Ru(dqpCO_2_Me)_2_(ptpy)^2+^ (DIP = 4,7-diphenyl-1,10-phenanthroline, bdt = 1,2-benzenedithiolate, dqpCO_2_Me = 4-methylcarboxy-2,6-di(quinolin-8-yl)pyridine) and ptpy = 4′-phenyl-2,2′:6′, 2″ -terpyridine) were tested with *S. aureus* and *E. coli* [109]. The complexes were found active against the Gram(+) strain and to a lesser extent against the Gram(−) strain, such difference in susceptibility being commonly reported in studies using other PSs [110] (Figure 2B). This observation was further detailed and rationalized by us when investigating a collection of 17 metal-bearing derivatives; two neutral Ru(II) complexes (Ru(phen)_2_Cl_2_ and Ru(phen-Fluorene)_2_Cl_2_) as well as a mono-cationic Ir(III) complex (Ir(phen-Fluorene)(ppy)^2+^; ppy = phenypyridyl ligand) were found almost inactive, whereas a dicationic Ru(II) (Ru(phen-Fluorene)(phen)_2_^2+^) was found to be the most active against a panel of clinical bacterial strains [11]. More recently, Sun et al. described a Ru(II) complex bearing photolabile ligands; they showed its ability to safely photoinactivate intracellular MRSA while inducing only negligible resistance after bacterial exposure for up to 700 generations [111]. Although the precise mechanism(s) of action is not well-established in every case, Ru(II/III) compounds are also increasingly considered for their potential anti-parasitic activity for combating neglected tropical diseases such as malaria, Chagas’ disease, and leishmaniasis. Moreover, potent antiviral activities have been noted for the ruthenium complex BOLD-100, particularly against HIV and SARS-CoV-2 [112]; importantly, this compound appears to retain its activity on all mutant strains of the SARS-CoV-2 [107]. All combined, metal complexes—especially ruthenium-based compounds—can display antimicrobial activity via multiple, likely synergistic, mechanisms, involving notably their ability to produce ROS. Therefore, they are of major interest for a wide range of aPDT-related applications.

### 4.2. Multicomponent PSs and Nanoscale Implementation: Extension to Nanoedifices with PSs

The eventual limitations in the use of singular photoactive molecules as PSs for aPDT applications reside notably in the recurrent lack of solubility and stability in the target media (typically leading to aggregates and/or PS quenching), biocompatibility (“immune stealth”) and bioavailability, but also in the relative absence of selectivity for a prospected target (e.g., efficiency in the interactions with a defined target, stimuli-responsive or alternate triggers for controlled release). Thus, the widespread nanoscale implementation into the development progress of upgraded PSs has indubitably provided significant flexibility to first address these drawbacks, and implicitly contribute to the optimization of the aPDT activity via an extensive panel of mainly exclusive features specific to nano entities; the latter typically include an advantageous surface to volume ratio (with a high PS per mass content), access to unique chemical/physical/biological properties (e.g., optical properties with QDs or magnetic ones with superparamagnetic iron oxide NPs (SPIONs)), and almost inexhaustible synthetic options in the design of nanoplatforms [113,114,115].

Due to their paramount structural diversity and intrinsic variety of properties, the classification of the so-called “PS nanosystems” or “nano-PSs”—which partly include and overlap with “conjugated systems” and “combinatorial strategies”—remains rather challenging; however, we can usually identify the following criteria as the main pillars to rationalize and compare these aPDT nanomaterials [27,80,116]:(i)Role(s) and nature of the nanocomponent(s) in the PS nanosystems: the two criteria typically considered for the discrimination of nano-PSs are the role and nature of the nano building block(s) involved. With regards to the role of the latter, we can conventionally discern on the one hand the “PS nanocarriers” (e.g., polymersomes or Au NPs) in which the nanomoiety acts as a delivery system for singular PS molecules (e.g., MB) while either complementing, facilitating, or enhancing the aPDT activity (depending on the nature and eventual intrinsic properties of the nanovector), and on the other hand, the “PS active” nanoagents with the nanocomponent endorsing the role of PS. Among the examples, some versatile nanotemplates may ultimately display a dual role, i.e., “active PS” and “PS conveyor” (e.g., ZnO NPs), while distinct nanomoieties might be simultaneously required for the design of utterly sophisticated hybrid nano-PSs (e.g., Au@AgNP@SiO_2_@PS) [117,118]. In addition to the chemical composition, the nature of the nano building block(s) will also be defined by the fundamental characteristics of nano-objects, such as size, shape, topology, and crystal structure, which will all ineluctably contribute to tailor the biological behavior of the nanomaterials and the interactions with the targets (e.g., with the membranes of the bacteria) [119]. Moreover, for the same nanocore, the nature and role of the eventual surfactant(s) involved (e.g., silica coating or poly(ethylene glycol) (PEG) coating for metal NPs) can drastically alter the overall behaviors of the nanosystems.(ii)Type of interactions between the nano entity and the PS, and localization of the PS in the nanosystems: other criteria of relevance when describing nano-PSs—specifically PS nanocarriers—reside in the nature of the interactions between the nanocomponent and the PS molecules involved, but also the location of the PSs. Thus, we can distinguish the common cases of PS molecules “embedded” within a nanovector either by physisorption or functionalized (chemisorption), and alternately the nanoplatforms with surfaces decorated with PSs, again, either by physical or chemical adsorption. The differences between the two types of interactions and distinct localizations of the PSs implicitly imply distinctive chemical engineering and related requirements, and may potentially impact the resulting stability of the nanoedifices, but also the aPDT activity. For instance, in the case of PS molecules located inside the edifice and not released, the selected “nanomatrix” should adequately permit the photoactivation process of the internalized PSs, be sufficiently porous/permeable to both triplet and singlet oxygen and eventual ROS generated by the photosensitizers (i.e., efficient internal diffusion of molecular oxygen to react with the PSs then external diffusion of ^1^O_2_ to the targets) while also presenting inertness to the latter to not compromise or quench the aPDT activity. Meanwhile, with surfaces of nano-objects decorated with PSs, the PSs may then contribute to some extent as an interface with the biological medium or the target.(iii)Biological impacts of the PS nanosystems: in addition to the biocompatibility and aPDT efficiency (including the critical concentrations just as the half-maximal effective concentration EC_50_, minimum inhibitory concentration MIC, or 50% growth inhibition concentration GIC_50_), the eventual biodegradability, elimination process, or ecotoxicity of the aPDT nanomaterials can markedly vary from one system to another (based on factors such as composition and size/shape), but are rather difficult to evaluate or compare; ergo, these factors are not systematically addressed in the reports.(iv)Relative sustainability of the nano-PSs for aPDT applications: the reproducibility, eco-friendliness, and cost-effectiveness parameters of the synthetic protocols and production of aPDT nanomaterials, as well as the ease of storage and use, and the stability over time are also ultimately to be evaluated for any system aiming to be viable and reasonably applied; however, similar to (iii), these parameters are complex and so scarcely investigated.

Thus, in the overview presentation of the different PS nanosystems hereafter, the chemical features (i) and (ii) have been conventionally defined as the main criteria for the classification. Alternatively, the nature of the aPDT applications has also been used as the main criterion for classification in some references [120]. Another approach consists of systemizing all the nanosystems dedicated to a given PS (e.g., curcumin) [121].

Within the extensive collection of aPDT nanomaterials reported to date, the majority belongs to the category labeled as “PS nanocarriers” with the nanocomponent acting as a delivery system for PS molecules; however, increasing examples involving PS-active nano building blocks have emerged as well. Overall, this multipurpose role of the nanomoiety may include avoiding aggregation (e.g., dimerization, trimerization) and correlated PS quenching, enhancing “solubility” (i.e., dispersibility), stability and bioavailability, allowing “biological stealth”, on-demand release and target specificity, and ultimately triggering eventual synergistic aPDT activity with the complementary or ameliorative intrinsic properties of the nanocomponent; although irrevocably confirmed in many nano-PSs, the mechanisms involved in the synergy may differ from one system to another, and often remain partly or integrally unresolved due to the complexity of these tacit multiparameter contextures [113]. It is noteworthy that most systems comprise “classic”/”traditional” organic PSs (natural or synthetic, e.g., curcumin, MB) with fewer examples involving metallated PS molecules such as the recent review from Jain et al. dedicated to ruthenium-based photoactive metalloantibiotics [108]. Among aPDT nanomaterials, we can thus identify various families of nanoplatforms based on the nature of the nanocore, starting here with the inorganic vectors followed by the organic templates; as an indication, in the common cases of “multi-component nano-PSs”, the classification has been defined hereafter according to the main/prominent nano building block involved in the composition, i.e., metal-based systems, silicon-based systems, carbon-based systems, lipid-based systems, and polymer-based systems. Regarding the following presentation of aPDT nanosystems, it is important to specify that it is not exhaustive, but instead provides a panorama of the main categories of nanosystems—either colloids or surfaces [27,122]—and their related specificities, with an emphasis on recent developments.

#### 4.2.1. Metal-Based Systems

Metal-based nanostructures have been extensively investigated—both as “PS cargo” and PS active entities—through the exploration of the richness and diversity of the respective subcategories related to this class of compounds, as detailed below. Each of the below-mentioned inorganic classes presents distinct specificities of relevance for aPDT applications, with a choice to be defined on a case-by-case basis according to the target, the nature of the PSs involved (with possible preferential affinity), or the anticipated complemental or synergistic role of the selected nano entity (based upon its chemical, physical, and/or biological features, with the eventual PS molecules located either at the surface of the NPs or within, when present). It is important to notice that the nanocores from each subcategory can be either further implemented, with silica or polymer coating for example, or combined (multicomponent nano-PSs) in order to adequately optimize the efficiency of the systems [117]; however, the outcomes of these hybridity processes are complex to anticipate with systematic rationality, with either enhancement or quenching of the properties observed depending on the composition of the combinations.

##### Metal NPs

Metal NPs—mainly gold, but also silver—maybe considered among the “gold standards” in nanomaterials through the history and expansion of nanosciences in terms of dedicated publications and vastness of related possible applications [123]. As already well documented for Au and Ag NPs, the reasons are numerous and reside notably in their relatively easy accessibility with low-cost and highly reproducible (large scale) biogenic and chemical synthetic routes, in addition to the flexibility to finely tailor the properties via a refined size and shape control (with narrow size distribution and diverse shapes), and the facility for functionalization with various types of molecules. Moreover, gold NPs display biocompatibility, low toxicity, and immunogenicity, almost chemical inertness (distinctly from their inherent catalytic properties), while silver nanomaterials present intrinsic antimicrobial activity against a broad spectrum of microorganisms and related MDR infections (e.g., towards Gram(−) and Gram(+) mature biofilms of MRSA), and disruption of biofilm formations while being safe for mammalian cells [124,125,126].Ultimately, both Au and Ag NPs share nano features specific to noble metal systems, i.e., localized surface plasmon resonance (LSPR, arising from their resonant oscillation of their free electrons upon light exposure) and resonance energy transfer (RET), with subsequent optical and photothermal properties of enhancing appositeness in aPDT applications and PDI efficacy (e.g., ROS production) [127,128,129]. For the most part, Au and Ag NPs of various shapes (e.g., spheres, rods, cubes) are combined with organic PS molecules such as RB and MB [128,129,130,131,132,133,134,135,136], but also with metallated PSs such as ruthenium complexes, metallophthalocyanines, and metalloporphyrins [108,137,138]; the corresponding PS nanovehicles can also be labeled as “conjugates” but they strictly differ from the “mixtures” involving metal NPs and PS molecules [139]. Other noble metal NPs, viz., platinum, have also been employed in aPDT applications due to their multitarget action to inactivate microbes, although to a lower degree up to now owing to synthetic limitations [27,140]. Alternately, redox-active copper NPs are typically less costly and easier to access and present unique features among which the faculty to generate oxidative stress to various microbes through the genesis of ROS [141], such as in the recently developed copper–cysteamine (Cu–Cy) nano-PS that can be activated either by UV, X-ray, microwave, or ultrasound, to produce ROS against cancer cells and bacteria [142,143]. More unwontedly, approaches to treat subcutaneous abscesses lead to the use of acetylcholine (Ach) ruthenium composite NPs (Ach@RuNPs) as an effective appealing PDT/PTT dual-modal phototherapeutic killing agent of pathogenic bacteria, with Ach playing a role in targeting the bacteria and promoting the entry into the bacterial cells [108]. While belonging to the same category, each metal displays particular specificities; consequently, with the objective of optimization, nano-PSs resulting from alloys or multimetallic NPs have been further designed such as the Au@AgNP@SiO_2_@PS and AA@Ru@HA-MoS_2_ (AA: ascorbic acid, HA: hyaluronic acid) nanocomposites [117,118].

##### Metal Oxides

Similar to gold and silver, metal oxides such as iron oxide, titanium oxide and zinc oxide have been cornerstone contributors in the global evolution of applied nanomaterials (particularly in medicine), due notably to intrinsic magnetic and optical nanoscale features, with the latter typically available at “room temperature” and commonly finely tunable via shape, size, and crystal structure parameters [144]. Indeed, specific single-domain superparamagnetic iron oxide NPs (SPIONs)—either magnetite (Fe_3_O_4_) or maghemite (γ-Fe_2_O_3_) of various shapes and sizes, including ferrite or doped derivatives—exhibit an outstanding magnetization behavior with no remanent or coercive responses upon exposure to a magnetic field. As a result, such magnetic NPs have legitimately generated interest and use as magnetic resonance imaging (MRI) and magnetic particle imaging (MPI) agents, as well as magnetic fluid hyperthermia (MFH), magnetic cell separators [145], or drug delivery conveyers with the possibility to guide the NPs to the targeted area via external magnetic fields [146]. More recently, iron oxide nano-objects proved to be also of pertinence for aPDT applications not only as a magnetic “nanocargo” for various organic and inorganic PSs such as curcumin, MB, ICG, BODIPYs, porphyrins, metallophtalocyanines, or ruthenium derivatives among others [27,108,147,148,149,150,151,152,153,154,155], but also with peroxidase-like activity to enhance the cleavage of biological macromolecules for biofilm elimination [156]. Extension in the design of more elaborate multicomponent architectures involving hybrid iron oxide nanocore led *inter alia* to Ag/Fe_3_O_4_, Ag/CuFe_2_O_4_, CoFe_2_O_4_, and Fe_3_O_4_/MnO_2_ NPs conjugated with different PSs [27,117,147,150,151,157,158,159,160].

Other oxides have also drawn heavy attention, in particular zinc oxide and titanium oxide, as the photophysical properties of these wide bandgap semiconducting nanomaterials efficiently translate into a multi-level antimicrobial activity including PS vessel and/or PS active agent (with possible coupled aPDT response), and/or membrane disruptor [161,162,163,164]. Thus, ZnO and TiO_2_ nanoplatforms possess the ability to alter microbes’ integrity—through alternate mechanisms involving ROS and/or metallic ions—in the dark or via photoactivation [165]. The latter is customarily triggered by UV or X-ray irradiation, with the eventual possibility to adequately shift to other wavelengths such as visible light irradiation in virtue of diverse modification methods of the oxides encompassing notably: doping or surface alteration (e.g., F-doped ZnO, coatings or oxygen deficiency), coupling with other bandgap semiconductors (e.g., ZnO/TiO_2_) or sensitizing dyes, and composites [121,156,164,166,167,168,169,170,171]. Furthermore, beside the size and shape of the NPs, the crystallographic phase appears to be a tuning parameter of particular importance for the antimicrobial effects of some oxides, especially for TiO_2_ with the distinction between the anatase, rutile, and brookite structures [172,173,174,175]. Although reported to a lesser extent, the list of alternate oxides exhibiting potential in aPDT applications comprises CuO/Cu_2_O, MnO_2_, and rare earth oxides to mention just a few [141,176,177,178,179,180].

##### QDs and Metal Chalcogenide Nanomaterials

Aside from zinc/titanium oxides, distinct semiconductors such as QDs and metal chalcogenide (e.g., metal sulfide) nanomaterials (involving elements from different groups in the periodic table) proved to be efficient disruptors against various multi-drug-resistant microorganisms. Due to their smaller size of a few nanometers (ca. up to 10 nm), QDs differ from other nano-objects with physical and optoelectronic properties governed by the rules of quantum mechanics, high chemical stability and resistance to photobleaching, and near-infrared (NIR) emission (typically above 700 nm) notably allowing for deep-tissue imaging [181]. Appositely comparable, metal chalcogenide likewise reveals unorthodox physio and physicochemical properties, accordingly garnering a legit interest for antimicrobial applications [182]. Consequently, not only can these nano building blocks carry PS molecules and alter the integrity of microbial walls/membranes or gene expression, but they may as well act as PSs; when coupled with other PSs, synergistic interactions in the QD-PS edifices might occur resulting from mechanisms such as Förster resonance energy transfer (FRET, non-radiative energy transfer from QD donors to PS acceptors) to generate free radicals and ROS. Among examples of such QDs and metal chalcogenide aPDT systems can be cited CdTe QDs and related CdTe-PS conjugates, CdSe/ZnS QDs combined with PSs, InP and InP-PS, Mn-doped ZnS, MoS_2_, but also CuS and CdS nanocrystals, with ultimately hybrid systems involving for instance CoZnO/MoS_2_ or AgBiS_2_–TiO_2_ composites [123,183,184,185,186,187,188,189,190,191,192,193,194,195,196].

##### Metal–Organic Framework (MOF) Nanoscaffolds, Upconversion Nanomaterials and Other Metal Ion Nanostructures

Although less investigated than the above-mentioned alternatives, other original metal-based nanostructures identified as MOF nanostructures, upconversion nanoplatforms, and alternate metal ion nanomaterials tend to further consolidate their promising potential for aPDT applications. Considering their towering surface area and porous ordered structure with substantial loading capacity (e.g., adsorption of O_2_ and ensuing photocatalytic production of ^1^O_2_ via a heterogeneous process), stable versions of colloidal nano-MOFs—or less common covalent organic frameworks (COFs), i.e., reticulation variants typically defined by non-metal “nodes” instead of metal ones—have emerged as efficient heterogeneous photosensitizers—with frameworks acting as PSs or entrapping PSs—towards antimicrobial applications (e.g., enhanced penetration for bacterial biofilms eradication), with porphyrin-based or porphyrin-containing MOFs and COFs, or Cu-based MOFs embedded with CuS NPs for rapid NIR sterilization among recently reported solutions [197,198,199,200,201,202,203,204,205].

Moreover, upconversion NPs (UCNPs) generally involve actinide- or lanthanide-doped transition metals and refer to the nonlinear process of photon upconversion, viz., a sequential absorption of two or more photons resulting in an anti-Stokes type emission (i.e., emission of light at a shorter wavelength than the excitation wavelength); when translated into biomedical context, UCNPs can be typically activated by NIR light—characterized by deeper tissue penetration and reduced autofluorescence, phototoxicity, and photodamage when compared with UV or blue light—and produce high energy photons for optical imaging or more recently aPDT when combined with PSs [206,207,208,209]. Intrinsically limited by low upconversion quantum yield, the current focus consists of developing hybrid UCNPs to improve aPDT efficiency; thus, auspicious progress has been achieved with examples such as {UCNPs (NaYF4:Mn/Yb/Er)/MB/CuS-chitosan)} multicomponent nanostructured system revealing a superior antibacterial activity with the UCNPs enhancing the energy transfer to MB, the CuS triggering synergistic PDT/PTT effects, and chitosan assuring stability and biocompatibility [156]. Other examples also include silica coating β-NaYF_4_:Yb, Er@NaYF_4_ UCNPs loaded with MB as PS and lysozyme as a natural protein-inducing bacterial autolysis, Fe_3_O_4_@NaGdF_4_:Yb:Er combined with the photo/sonosensitizer hematoporphyrin monomethyl ether (HMME), UCNPs@TiO_2_, N-octyl chitosan (OC) coated UCNP loaded with the photosensitizer zinc phthalocyanine (OC-UCNP-ZnPc), or the UCNPs-CPZ-PVP system (CPZ: β-carboxy-phthalocyanine zinc, PVP: polyvinylpyrrolidone) to name but a few [206,210,211,212,213].

Alternatively, more disparate metal-ion aPDT systems have been reported such as PS encapsulated dual-functional metallocatanionic vesicles against drug-resistant bacteria involving copper-based cationic metallosurfactant, or self-assembled porphyrin nanoparticle PSs ZnTPyP@NO using zinc meso-tetra(4-pyridyl)porphyrin (ZnTPyP) and nitric oxide (NO) [214,215,216].

#### 4.2.2. Silicon-Based NPs

This category will be divided hereafter into two main subclasses—porous silicon (pSi) and (mesoporous) silica (SiO_2_)—which differ in the oxidation state of the silicon and display distinctive properties, more specifically different quenching behavior and photodynamic activity (singlet oxygen quantum yield under irradiation).

Porous silicon NPs (pSi NPs) are among the most promising types of inorganic nanocarriers for biomedical applications and have been intensively investigated since the first publication by Sailor et al. in 2009 regarding their application for in vivo treatment of ovarian cancer. Composed of pure silicon, pSi NPs indeed exhibit relevant features encompassing not only pores with large capacity for drug loading combined with specific surface area allowing for implemental functionalization, but also degradability in an aqueous environment, and biocompatibility [217]. Moreover, porous silicon particles are known to be photodynamically active with related inherent antimicrobial properties by generation of ROS under irradiation with light of a specific wavelength [218,219]. Because of the low quantum yield of singlet oxygen production from porous silicon itself, particles can be grafted with additional PSs, such as porphyrins, to enhance the yield of singlet oxygen generation and thereby antimicrobial properties for PDT applications [219,220]. Consequently, several silicon-based systems have been reported in recent years, mainly for PDT applications [221,222]. Furthermore, pSi NPs display intrinsic fluorescence, which can be applied for imaging and real-time diagnostics regardless of any surface functionalization [220,223].

As previously mentioned, pSi has a low singlet oxygen quantum yield due to quenching, which makes silica particles in comparison a more suitable substitutional system combining similar biocompatibility with improved optical properties. On the other hand, one of the pivotal advantages to use silica (SiO_2_) conjugates with PSs is to achieve a better “solubilization”—or more accurately, dispersibility—of hydrophobic dyes and a better photostabilization, thus limiting the self-photobleaching of PSs. Further advantages to name as the most important features of silica are high biocompatibility, antimicrobial properties, and high surface area for mesoporous silica that can be synthesized easily from commercially available precursors [27,224,225]. Furthermore, SiO_2_ exhibits an effective PS-grafting capacity [226]; the latter can be accomplished via adsorption, covalent bonding, binding to the hydroxyl groups from silica surface, and entrapment during formation in silica particles or matrix [27]. Recently, Dube et al. reported about the photo-physicochemical behavior of silica NPs with (3-aminopropyl)triethoxysilane (APTES), and subsequently PS-modified surfaces for aPDT [150]. In addition, silica coatings have also been reported to prevent the degradation of nanocarriers (magnetite) and prolong the stability and functionality of PS systems [227]. Interestingly, coupling PSs to silica or Merrifield resin leads to distinct advantages; indeed, immobilized Ce6 notably displays significantly higher aPDT efficiency in comparison with the free form, which is probably due to an enhancement of the adhesion of PSs to bacterial cells resulting in a stronger cell wall disorganization [228,229]. Unsurprisingly, many approaches with encapsulated PSs in silica NPs for potential aPDT applications have been reported in recent years [229,230,231,232].

Distinctly, combinatory approaches involving other nanocomponents, such as silica-containing core-shell particles or silica-coated inorganic NPs, have emerged to further implement the properties of silica with the specific features (e.g., magnetic, photoactive, or antimicrobial properties) from other nanomaterials of relevance [118]. Thus, since the surface of silica can be easily grafted with PSs and is highly biocompatible, the surface modification of silica particles using metallic NPs (e.g., Ag NPs) to enhance the antibacterial photodynamic activity, has been developed for improved aPDT [233] while combinations with carbon quantum dots have been assessed for imaging-guided aPDT [234]. Another example refers to sufficient aPDT/PDI systems, more precisely to mesoporous silica-coated NaYF4:Yb:Er NPs with the PSs (silicon 2,9,16,23-tetra-tert-butyl-29H,31H-phthalocyanine dihydroxide) loaded in the silica shell to enhance bacterial targeting of *E. coli* and *S. aureus* [235].

In addition to silica-based NPs or silica-containing core-shell particles, lesser-known silica nanofibers also proved to be suitable substrates for potential aPDT, PDT, and PDI applications. As an illustration, Mapukata et al. [236] recently reported silver NP-modified silica nanofibers with embedded zinc phthalocyanine as PS for aPDT applications. The nanofiber-based substrates offer the advantage of fast removal after application, which can allow limiting any dark toxicity [236,237,238].

Silica NPs and fibrous or dendritic fibrous nanosilica have also been reported for the formation of nanocomposites to create antimicrobial photodynamically active surfaces for aPDT or PDI; to create such surfaces, silica NPs can be embedded into polymeric matrices for enhanced biocompatibility and complementary surface properties from the selected polymer [239].

Additionally, silica substrates and nanoconjugates offer a suitable platform for combinatory approaches since they can be easily modified. For example, Zhao et al. described polyelectrolyte-coated silica NPs modified with Ce6 [240]. These complexes could be extracted, by bacteria, from silica NPs to form stable binding on the bacterial surface, changing the aggregation state of Ce6 and leading to both the recovery of PS fluorescence and ^1^O_2_ generation. Such bacteria-responsive multifunctional nanomaterials allowed for simultaneous sensing and treating of MRSA. Another approach is illustrated by the photo-induced antibacterial activity of amino- and mannose-decorated silica NPs loaded with MB against *E. coli* and *P. aeruginosa* strains [241]. The modification of silica substrates with mannose led to an increased targeting of *P. aeruginosa* and reduced dark toxicity of the systems.

#### 4.2.3. Carbon-Based Nanomaterials

Akin to silicon-based nano-objects, the notable diversity of allotropic customizable carbon-based nanostructures—either conveyers for traditional PS molecules or intrinsically PS active—legitimizes their distinct consideration in the actual classification of nano-PSs [242,243,244], with the following subcategories.

##### Fullerenes, Carbon Nanotubes (CNTs), and Nanodiamonds

In the evolution of carbon-based nano-objects, fullerenes and carbon nanotubes derivatives may be chronologically introduced as the “first generation”. Discovered in the mid-1980s, the proper C_n_ (*n* = 60–100) spheroidal “soccer ball” π-conjugated structures of the fullerenes yield tremendous chemical modularity and electrochemical and physical properties, including photostability, the propensity to act as a PS via Type I or Type II pathways (Figure 1A) with high ROS quantum yield, and oxygen-independent photo-killing by electron transfer. Despite their intrinsic hydrophobicity typically requiring surface functionalization for biocompatibility and related dispersibility, they have proven even nowadays their effectiveness as broad-spectrum photodynamic antimicrobial agents, with photoactive antimicrobial coating based on a PEDOT-fullerene C_60_ polymeric dyad (PEDOT: poly(3,4-ethylenedioxythiophene)), BODIPY-fullerene C_60_, diketopyrrolopyrrole–fullerene C_60_, and cationic fullerene derivatives among recent examples [80,92,156,245,246,247,248,249,250,251]. Mainly developed a few years later, carbon nanotubes (CNTs)—either single wall CNTs (SWCNTs) describable simply as a single-layer sheet of a hexagonal arrangement of hybridized carbon atoms (graphene) rolled up into a hollow cylindrical nanostructure, or multiwall CNTs (MWCNTs) consisting of nested SWCNTs—unveil both independent capacities to produce ROS upon irradiation and high surface area for decoration with PS molecules [156]. Neoteric specimen of PS-CNTs encompass toluidine blue, polypyrrole, malachite green, MB, RB, and porphyrins [156,252,253,254,255,256,257]. Although purportedly older since it was discovered in the 1960s, diamond NPs or nanodiamonds seemingly remain the lesser known carbon-based nanomaterials to date; nevertheless, the latter dispose of legit aPDT arguments with their fluorescence, photostability, proclivity for conjugation with diverse PSs such as porphyrins or metallated phthalocyanines and silver NPs, but also inherent antibacterial activity [120,258,259,260,261,262].

##### Carbon QDs (CQDs)

As the next momentum in the blossoming of “nano-carbon” era, the carbon QDs were discovered in the early 2000s [263,264]. The physical and chemical properties of these fluorescent particles, commonly quasi-spherical with less than 10 nm in diameter, can be finely tuned upon size/shape variations or doping with heteroatoms (e.g., B, N, O, P, S) [263,265]. By virtue of their biocompatibility and dispersibility, photostability, low toxicity and related eco-friendliness, good quantum yield and conductivity, CQDs have been investigated for various applications, and more recently as antimicrobial agents; withal, their environment-friendly features combined with low cost and rather ecological biogenic or synthetic routes (from natural or synthetic precursors) place them advantageously as a viable scalable photocatalytic disinfection material compared with alternate nano-PSs. Late cases involve doped or hybrid CQDs or more conventional conjugates of CQDs with PSs [121,266,267,268,269,270,271].

##### Graphene, Graphene QDs (GQDs), and Graphene Oxide (GO) Nanostructures

In a similar timescale to CQDs is the quantum leap discovery and blooming of graphene and graphene oxide materials. Graphene can be defined as a 2D allotrope of carbon, more accurately a monolayer of atoms with a hexagonal lattice structure (or single-layered graphite) and identifiable as the “building block” for the discrete fullerenes, 1D carbon nanotubes, and 3D graphite. Despite its stunning mechanical/electronic properties and chemical inertness, the limitations of graphene, such as zero bandgap and low absorptivity, lead to the ulterior conversion of the 2D graphene into “0D” GQDs [272,273]. Due to quantum confinement and edge effects, GQDs exhibit different chemical and physical properties when compared with other carbon-based materials, as well as a non-zero bandgap, good dispersibility, and propensity for functionalization and doping. Structurally, GQDs differ from CQDs because they comprise graphene nanosheets with a plane size less than 100 nm [272,274]. Likewise, graphene oxide (GO) is the oxidized form of graphene i.e., a single atomic sheet of graphite with various oxygen-containing moieties either on the basal plane or at the edges. Meanwhile, reduced graphene oxide (rGO) can be summarily described as an “intermediate” structure between graphene and GO, with variable and higher C/O elemental ratios compared with GO, but remaining residual oxygen and structural defects with reference to the pristine graphene structure. Although GO was reported a couple of centuries ago, GO and rGO nanomaterials have mainly emerged for various applications after the discovery of graphene since GO is a precursor to prepare graphene, and both present distinctive physical and chemical properties that differ from graphene. As a result, countless and rising fast illustrations of graphene derivatives for antibacterial applications are regularly reported [121,275,276,277,278,279,280].

#### 4.2.4. Lipid-Based Systems

Due to their amphiphilic nature (typically hydrophilic “head” and hydrophobic “tail”), some lipids—natural and synthetic—have been extensively studied to develop efficient biocompatible delivery systems—initially for drugs and DNA/RNA, but also for aPDT PSs—with synthetic flexibility and structural diversity. Among the prevalent examples, we can distinguish the micelles (lipid monolayers with polar units at the surface and hydrophobic core) and the liposomes (one or more concentric lipid bilayer with a hydrophilic surface and an internal aqueous compartment). Although sharing similar chemical constituents, the micelles and liposomes present significant differences to be taken into consideration depending on the intended application (nature of the target) and the nature of the PSs. Indeed, the micelles are typically smaller than liposomes (with a diameter starting from a few nanometers for the micelles, and ca. 20 nm for the liposomes), with distinct stability and permeability in biological medium and uptake pathways for the PSs into bacteria. With reference to the nature of the transported PSs, the liposomes display the additional flexibility to carry both hydrophilic PSs (in the core compartment or between the bilayers) or/and hydrophobic PSs (within the lipid bilayer), while the micelles are usually easier and cheaper to prepare [80,156]. As often critical to address for biomedical applications, the surface charge of these nano-objects can be tailored to further optimize the interactions with the bacteria, with cationic modification of liposomes identified as a promising aPDT efficiency “amplifier” [80,156,281]. In addition to recent examples such as the hypericin loaded liposomes against Gram(+) bacteria [282,283,284,285,286], another emerging and promising alternative includes the development of modified liposome-like derivatives labeled either as “ethosomes”, “transfersomes”, or “invasomes”, which can be briefly described as ultra-deformable vesicular carriers with upgraded transdermal penetration and increased permeability into the skin for the PSs compared with conventional liposomes [287,288,289,290]. On the other hand, recently reported aPDT micellar systems refer to micelles loaded with various hydrophobic PSs such as curcumin, BODIPY, porphyrins, hypocrellin A, or hypericin among others [121,291,292,293]. Furthermore, solid lipid NPs (SLN) composed of solid biodegradable lipids have been recently highlighted as delivery systems used for actual mRNA COVID-19 vaccines [294], but they also have been reported as transporters for curcumin for the treatment of oral mucosal infection [121]. Besides, we may also include nanoemulsions in this category of lipid-based nanovehicles since the latter conventionally involve lipids in the oily phase during the formulation process, with recent curcumin/curcuminoid nanoemulsions [121,295].

#### 4.2.5. Polymer-Based Systems

In direct correlation with the above-mentioned lipid-based systems, polymer-based nanocarriers have been positioned as a logical extension with the objective to implement a “degree of freedom” in the synthesis flexibility while expanding the panel of building blocks available in the design of aPDT nanostructures. A categorization of polymeric systems for the delivery of PSs to therapeutically relevant sites can be done using as criteria either the nature of the polymer(s) involved or the type of polymeric (nano)structures. Thus, in the following, we will use a classification primarily based on the structure of the polymeric systems with differentiation between NPs (including hydrogels, biopolymers, and aerogels), polymersomes, polymeric micelles, dendrimers, fibers, and polymeric films and layers (including hybrid systems and nanocomposites). The nanostructure of polymeric systems is implicitly highly correlated to the molecular structure, i.e., composition (nature of the polymer(s), ratios, and distribution and amount of hydrophilic and hydrophobic moieties), charge, and size, as reviewed recently by Osorno et al. [296]. The refined control of these parameters enables the design and fine-tuning of specific polymeric nanostructures spanning from long-ranged ordered lamellar sheets, tubes, and fibers to oval or spherical particles, micelles, and polymersomes [296] (Figure 2C).

##### Conjugated Polymers as PSs or Polymer-Functionalized PSs

One of the easiest approaches reported to develop polymeric carrier systems is to enhance water solubility and biocompatibility of PS molecules through the use of functionalized polymers, by introducing the PSs in a post-modification reaction to a hydrophilic and/or biocompatible polymer, or to modify the PS with a polymerizable group (e.g., acrylate) to react as a monomer for further polymerization with suitable monomers. Other apt options are conjugated polymers incorporating a backbone with alternating double and single bonds which provide photodynamic and optical properties, and therefore might act as PSs themselves [297,298]. For example, poly[(9,9-bis{6′-[N-(triethylene glycol methyl ether)-di(1H-imidazolium)-methane]hexyl}-2,7-fluorene)-co-4,7-di-2-thienyl-2,1,3-benzo-thiadiazole] tetrabromide (PFDBT-BIMEG) is a conjugated polymer which affords salt bridges and electrostatic interactions with microorganisms; these interactions enable the simultaneous detection and inhibition of microorganisms [297]. Conjugated polymers were intensively investigated for the development of multifunctional NPs in antibacterial applications because of their generally low-toxicity toward eukaryotic cells, their flexibility, and their high potential to vehicle versatile therapeutic molecules [298]. Improved PSs based on conjugated polymers or polymerized PSs with antimicrobial photodynamic properties have been amply reported in recent years [299,300,301,302,303]. For example, Huang et al. demonstrated the efficiency and selectivity of polyethyleneimine-Ce6 conjugated in potential aPDT/PDI applications [304].

Other polymeric systems also show antimicrobial properties or preferential target infection sites and are thus suitable for aPDT applications when combined with PSs. For example, cationic polymers, which are known to be highly hydrophilic, can be used to target cell walls and thus microbial infections. Hence, such polymers are adequate for potential aPDT applications [305,306]. Exemplary amphiphilic or cationic poly(oxanorbornene)s doped with PSs, which exhibit pronounced antimicrobial activity (99.9999% efficiency) against *E. coli* and *S. aureus* strains, have been reported [307]. By contrast, anionic polymer particles and nanocarriers with negatively charged surfaces and membranes invade the reticuloendothelial system, which leads to elongated blood circulation times [308,309,310]. Polymers with large fractions of functional groups, such as the biopolymers mentioned above, can be easily modified and offer many loading sites for suitable PSs. Thus, among many suitable materials, hyperbranched and dendrimeric polymers are particularly promising candidates for PS nanocarriers.

##### Dendrimers

Dendrimers, dendrimeric polymers, or dendrons are highly ordered and highly branched polymers, which may form spherical three-dimensional structures with diameters typically ranging from 1–10 nm [311]. The dimensions of dendrimers are relatively small compared with other drug delivery systems, but as they consist of individual well-defined molecules, the drug loading can be subtly established with reproducibility. The incorporation or encapsulation of drugs may be achieved by covalent binding of the PSs or drug molecules to reactive functional groups along the dendrimeric structure [312,313]. Alternatively, the PS or drug molecule may act as a scaffold from which the dendrimer is synthesized. Finally, drug or PS encapsulation may occur inside the voids of the dendrimer [311,314,315]. Thus, dendrimers offer a substrate for PSs with many reaction sites, and controllable sterical and hydrophobic properties depending on the backbone of the dendrimer. Approaches using hyperbranched polymers for stimuli-responsive release of PSs, such as porphyrins, under acidic or reductive conditions, with improved targeting of bacterial sites have been reported, such as mannose-functionalized polymers by Staegemann et al. [316,317].

##### Polymeric NPs and Nanocomposites

In many cases, polymeric nano-systems are not clearly classified, but instead typically grouped under the terminology “polymeric NPs”, since the exact structure may not be fully characterized or considered less relevant with respect to the application efficiency. Polymeric NPs are commonly defined as physically or chemically crosslinked polymer networks with a size in the range of 1 to 1000 nm, but if not further defined may also include nanocapsules, such as polymersomes, micelles, chitosomes, and even highly branched polymers and dendrimers [318].

In aPDT, polymeric NPs may be either used to encapsulate PS molecules (loading with PSs), or built from inherent photoactive polymers acting as PSs [303]. Various PS molecules (e.g., RB, porphyrin derivatives, or curcumin) encapsulated in biocompatible polymeric NPs, such as polystyrene-, PEG-, polyester- (including poly-beta-amino esters), or polyacrylamide-based NPs, are widely used in aPDT/PDI approaches, as recently reported [319,320,321]. In addition to the role of nanovehicle, the polymeric particles may also help to address the lack of solubility in the biological medium from the PSs and/or reduce their toxicity, as reported among others by Gualdesi et al. [320,322]. For instance, polystyrene NPs with encapsulated hydrophobic TPP-NP(5,10,15,20-tetraphenylporphyrin) have been reported as nano-PS for efficient aPDI approaches towards multi-resistant bacteria [321]. In addition, biocompatible PLGA (polylactic-co-glycolic acid) NPs were employed to encapsulate curcumin, which could potentially serve as an orthodontic adhesive antimicrobial additive [322]. Alternatively, combinatory approaches such as polymeric NPs merging a polymeric PS, a photothermal polymeric agent, and poly(styrene-co-maleic anhydride) as dispersants have recently been reported for coupled aPDT/PTT nano-platforms in aqueous media [323,324]. Furthermore, Kubát et al. reported an increase in the stability of physically crosslinked polymeric NPs (polystyrene) to comparable nanocapsules (PEG- poly(ε-caprolactone) (PCL) micelles) equipped with identical PS [319].

The term polymeric NP also includes NPs obtained from natural polymers (e.g., chitosan and alginate), hydrogels, or aerogels. Hydrogels are chemically or physically crosslinked polymeric networks, which are able to absorb large amounts of water due to their pronounced hydrophilicity, but do not dissolve because of the crosslinking, whereas aerogels are formed by replacement of liquid with a gas resulting in low-density polymeric structures [325]. Recently, Kirar et al. reported PS-loaded biodegradable NPs fabricated from gelatin, a naturally occurring biopolymer, and hydrogel, for application in aPDT [326]. Moreover, also recently, some polymers such as Carbopol-forming hydrogel matrix entrapping PS molecules attracted attention because of their bio/muco-adhesive property, allowing a prolonged local PS delivery. Nevertheless, the viscosity of such systems can prevent the efficiency of PDT by decreasing the photostability and ROS production, thus calling for further optimizations [327,328]. Furthermore, the incorporation of hydrophobic PSs such as curcumin in polyurethane hydrogel has demonstrated to be an efficient PS release system [329].

Another suitable hydrogel for aPDT is chitosan, which is a polycationic biopolymer with good biocompatibility and antibacterial properties [330,331]. Indeed, cationic polymers, which can be antimicrobials by themselves such as chitosan and poly-Lysine, can assure a better recognition toward bacteria thus improving antimicrobial efficiency. Typically, polycationic chitosan may form nanogels or NPs in the presence of (poly)anionic molecules. Alternatively, another promising aPDT approach for the treatment of *Aggregatibacter actinomycetemcomitans* was recently reported [332,333], in which anionic PS indocyanine green was used to form PS-doped chitosan NPs. These NPs were shown to significantly reduce biofilm growth-related gene expression. Other similar approaches have also been reported, in which PS doped chitosan NPs showed enhanced cellular uptake and improved antimicrobial properties compared with free PS against different bacteria [212,334,335].

The above-mentioned approach has not only been implemented with NPs, but also in thin films and layers to create biocompatible and antimicrobial surfaces [336]. Indeed, in aPDT and especially aPDI, polymers are also used to create antimicrobial surfaces or matrices to embed PS units. Thus, combinatory approaches—using cellulose derivatives, alginates, chitosan, and other polymer-based materials as biocompatible substrates for PSs and nanocomposites to create photoactive antimicrobial surfaces—have recently been reported [337,338,339]. The development of such surfaces, and in particular antimicrobial membranes, is of considerable interest, especially in numerous and diverse fields in which providing hygienic and sterile surfaces is essential. As an illustration, Müller et al. reported polyethersulfone membranes doped with polycationic PS that provide antimicrobial properties for potential use as filter membranes in water purification or medicine [340]. Other recent distinctive systems for aPDI refer to self-sterilizing and photoactive antimicrobial surfaces made from (i) natural polymers such as chitosan doped with chlorophyll [336], (ii) “bioplastic” poly(lactic acid) surfaces coated with a BODIPY PS [341], or (iii) synthetic polyurethanes doped with curcumin and cationic bacterial biocides [336,342,343,344,345].

Alternatively, other combinatory approaches—in which polymeric nanocomposites embedding inorganic nanomaterials with relevant complementary features are conceived—have garnered attention in recent years. Examples include fullerenes or silver NPs that are incorporated as PSs into polymeric matrices [346], phthalocyanine-silver nanoprism conjugates [347], or mesoporous silica NPs loaded into polymer membranes [348]. Additionally, the embedding of zinc-based PS—such as Zn (II) porphyrin [349]—into polymer matrices to create antimicrobial polymers and polymeric surfaces for aPDI and possibly aPDT has also been reported, with pronounced antimicrobial activity against several bacteria strains and viruses [350,351]. Furthermore, protein-based approaches were developed as reported by Ambrósio et al. [352] and Silva et al. [353] using BSA (bovine serum albumin) NPs or BSA conjugated to PSs to improve solubility and biocompatibility.

Lastly, stimuli-responsive polymeric NPs are generally known to be sensitive to an internal or external stimulus (e.g., pH, temperature, light) and so of utmost interest for the controlled release of PSs for PDT and aPDT, whereas the stimuli-responsiveness depends on the structure and properties of the used polymer such as the assembly of polymer chains and linkages [354]. For example, Dolanský et al. reported light- and temperature-triggered ROS and NO release from polystyrene NPs for combinatory aPDT/PTT approaches [355]. Unlike micelles or polymersomes, crosslinked NPs have been reported to be thermodynamically stable, while stimulus-responsive behavior such as pH-responsive release of PSs has been more often achieved using self-assembled micellar or vesicular structures [319,356,357,358].

##### Polymersomes

Polymersomes are polymeric vesicles that resemble liposomes, which were previously described in Section 4.2.4 dedicated to lipid-based systems. Polymersomes are formed from amphiphilic copolymers, which self-assemble in aqueous media, resulting in capsules. The lumen of the polymersomes is filled with an aqueous medium and the wall comprises a hydrophobic interior with a hydrophilic corona on both inner and outer interfaces. Polymersomes typically form when the weight fraction of the hydrophilic parts (e.g., PEG in a PEG-b-poly(lactic acid) (PLA) block copolymers) comprises up to 20–40% of the polymer. They also form at comparatively large water fractions in the solvent shift method [359,360,361]. At higher weight fractions, the system tends to form micellar structures [296,362,363]. The hydrophilic and hydrophobic compartments inside the polymersomes enable the uptake and encapsulation of both hydrophobic and hydrophilic molecules [364,365,366,367], making polymersomes a relevant candidate for the development of advanced nanocarriers for PSs [368]. In aPDT and similar approaches, efficient delivery of the PSs to the targeted tissue is essential, not only to minimize toxic side effects and overcome low solubility in body fluids, but also to enable elongated circulation in the blood stream and prevent dimerization and quenching of the PSs. Li et al. reported that PSs encapsulated in polymeric nanocarriers exhibit an increased singlet oxygen ^1^O_2_ quantum yield compared with non-encapsulated PSs [308]; by contrast, non-encapsulated PSs may tend to aggregate and lose efficiency [308,369,370]. Additionally, most polymersomes offer certain advantages compared with the established liposome-based systems such as enhanced biocompatibility, lower immune response, controlled membrane properties, stimuli-responsive drug release, biodegradability, and higher stability, with those mainly resulting from the individual design of polymers and polymersomes for the anticipated applications [296,371].

The encapsulation of PSs into specifically designed nanosized polymersomes with stimulus-triggered release of PSs has been reported in recent years, including temperature, pH, and light-induced release of the PSs [296,372,373]. For instance, Lanzilotto et al. recently reported a system used for aPDT consisting of polymersomes of the tri-block-copolymer poly(2-methyl-2-oxazoline)-block-poly-(dimethylsiloxane)-block-poly(2-methyl-2-oxazoline) (PMOXA34–PDMS6–PMOXA34) encapsulating water-soluble porphyrin derivatives [372].

##### Polymeric Micelles

In contrast to polymersomes or liposomes, micelles are particles that contain a hydrophobic core. Typical micelles range between 10 to 100 nm in size. More specifically, polymeric micelles are formed of amphiphilic polymers, with a higher ratio of hydrophilic parts in the case of block copolymers compared with polymersomes [296,362,363]. They also form at comparatively small water fractions in the solvent shift method [359,360,361]. Due to their structures, micelles are implicitly used to encapsulate hydrophobic drugs or agents, such as PSs, to convey the latter to the target (e.g., cancer cells or microbial infections) while overcoming their low solubility in aqueous media [374]. Additionally, encapsulation may reduce the toxicity of the PSs [375,376]. When compared with polymersomes, one disadvantage is the lack of flexibility to encapsulate or transport both hydrophilic and hydrophobic molecules. On the other hand, as reported in the review by Kashef et al. [377], polymeric micelles are easier to produce, hence they are more cost-efficient than liposomes and potentially polymersomes, while providing similar applicable features [378,379]. Among examples, an aPDT system of polymeric micelles, fabricated from methoxy-PEG and PCL and loaded with the hydrophobic PS curcumin and ketoconazole for the PDI of fungal biofilms, has been reported with an increased water solubility and controlled release of the PSs on display [380]. Additionally, Caruso et al. recently reported the synthesis of thermodynamically stable PEG-PLA micelles for efficient aPDI of *S. aureus*, suggesting that this delivery system is promising in aPDI applications, which also reduces toxicity compared with pure PSs [291]. Moreover, a significant advantage of polymeric micelles compared with lipid-based micelles resides notably in the adjustability of properties by individual design of the polymer and membrane surface with respect to the selected application. Hence, the polymeric micelles can be equipped with target ligands and/or their morphology can be tuned to increase the cellular uptake of the micelle, PS, or other therapeutic agents in a specific tissue, such as in tumors [381]. Like polymersomes, polymeric micelles can be tuned to release drugs or collapse by application of external stimuli, such as light, temperature or pH [357,358].

##### Niosomes

Niosomes are closely related to liposomes. They are vesicular structures consisting of non-ionic surfactants, including polymers, and lipids such as cholesterol. The addition of lipids to the non-ionic surfactants leads to increased rigidity of the membrane and the vesicular structure. Niosomes range from 10 to 3000 nm in size, and also may include multi-layered systems typically consisting of more than one bilayer. Niosomes offer enhanced stability and biocompatibility compared with vesicular structures based on ionic surfactants [288]. Due to the bilayer structure and adjustability of the selected polymers for the applications of choice, they exhibit similar properties and offer similar advantages as polymersomes, and can be used for encapsulation of a variety of hydrophobic and hydrophilic molecules [288,358]. They may also be equipped with targeting ligands, such as folic acid that enhances uptake into cancer cells for PDT of cervical cancer [382]. A niosome-based system, using MB as encapsulated PS, has been reported for aPDT treatment of hidradenitis suppurativa [383].

##### Polymeric Fibers

Another approach to obtain PS carrier systems relies on polymeric or polymer-hybrid fibers with diameters in micro and nano range [384,385,386]. The fibers can either be loaded after formation with various PSs, such as cationic yellow or RB [387] incorporated into wool/acrylic blended fabrics to obtain antimicrobial properties, or they can be electrospun from PS containing polymer solution to form polymeric fibers loaded with PSs [388]. Since many polymers (such as nylon, cellulose acetate, or polyacrylonitrile) are used in the textile industry, those materials may offer a novel approach to create substrates or textiles with self-sterilizing and antimicrobial properties in the presence of visible light [386,389]. Furthermore, combinatory approaches using various NPs, such as magnetite NPs, to form polymeric fiber-based nanocomposites have shown some efficiency for antimicrobial photodynamic chemotherapy [390,391].

To conclude this Section 4.2, the list of above-mentioned aPDT nanosystems is non exhaustive and aims at providing an overview of the diversity and richness in the composition of aPDT nanomaterials. The next section focusing on “combinatorial strategies” partly overlaps with the description of aPDT systems, which further complicates the classification process. Moreover, a distinction must be settled between strict “mixtures” of components and “chemical combinations” of components in the development of aPDT treatments with possible synergistic behaviors.

## 5. Focus on Combinatory aPDT Approaches

Given its intrinsic characteristics, PDT is highly amenable to versatile combinations with other drugs, treatments, or modalities in view to potentiate therapeutic effects, which include enhanced efficacy, limitation of side effects, and reduction in the risk of resistance emergence. Proof-of-concept for various combinatory aPDT are thus being increasingly recorded, exploiting the additive/synergistic effects arising from single or multiple therapeutic species acting via different mechanistic pathways. The following part aims to review recent works following such strategies (Table 2).

### 5.1. “Basic” aPDT Combinations

#### 5.1.1. Combination of Several PSs

The simplest combinatory aPDT approach probably consists in combining two or more PSs in a single treatment. Thus, besides aPDT making use of a single PS, aPDT may rely on the simultaneous use of several PSs in an attempt to obtain additive or synergistic antimicrobial effects. The PSs combined may exhibit different photophysical characters showing complementarities. For example, carboxypterin-based aPDT upon sunlight irradiation demonstrates a significant planktonic bacterial load reduction [419]. However, eradication of biofilm formation needs a PS concentration 500 times higher than assays performed with planktonic forms. When combined with MB, Tosato et al. showed that reasonable concentrations of both PSs exert synergistic effect on both biofilm and planktonic MDR bacteria [411]. In other studies, alternative dual-PSs aPDT systems have shown even better antibacterial and antibiofilm properties [35,412].

#### 5.1.2. Addition of Inorganic Salts

The combination of PSs with inorganic salts can modulate the PDT effectiveness, potentiating or inhibiting the antimicrobial activity through the production of additional reactive species or quenching ^1^O_2_ [420]. As an example, azide sodium can modulate aPDT effectiveness by promoting or inhibiting the binding of bacteria with PSs, depending on lipophilicity of the latter. Indeed, azide sodium is a ^1^O_2_ quencher, but can also produce highly reactive azide radicals, via electron transfer from PS at the excited state. This has been reported with many phenothiazinium dyes and also fullerenes [93,421,422]. Other salts can amplify the bacterial killing mediated by MB-PDT such as potassium iodide (KI), a very versatile salt, involved in the generation of short-lived reactive iodine radicals (I^•^/I_2_^•−^) [423,424]; similar findings have been obtained when using KI with cationic BODIPY derivatives [425] or porphyrin–Schiff base conjugates bearing basic amino groups [426]. Potassium thiocyanate and potassium selenocyanate could also form reactive species, such as the sulfur trioxide radical anion and selenocyanogen (SeCN)_2_, respectively [427]. In addition, interactions between PSs and target microorganisms could be improved thanks to inorganic salts. For example, calcium and magnesium cations can modulate the electrostatic interaction between PSs and bacterial membranes [428]. It is worth noting here that most of the above-mentioned studies were conducted under in vitro settings. While inorganic salts can be useful to enhance in vitro or ex vivo aPDT effects, the use of such additives in animals or human beings would need to be carefully examined, considering the dose to be used and potential concomitant side effects.

### 5.2. Combinations of aPDT with Other Antimicrobial Drugs or Antimicrobial Therapies

Various classes of antimicrobial drugs or other antimicrobial therapies, which action does not depend on light, have been considered with regard to their aPDT compatibility.

#### 5.2.1. Antibiotics

Antibiotherapy is an obvious complementary therapy to aPDT, which may allow obtaining stronger antimicrobial effects and/or restore antibiotic susceptibility. Combinations of PSs with antibiotics have been investigated in numerous studies addressing various infectious diseases, such as skin and mucosal infections as recently reviewed [429]. Aminosides are the most common antibiotics used in combination with a PS. For instance, kanamycin, tobramycin, and gentamicin combined respectively to RB, porphyrin, and 5-Alanine have been reported, showing potent effects against bacteria of clinical interest, notably by improving biofilm clearance and reducing microbial loads [392,430,431,432,433]. In addition, combinations with other antibiotic classes, such as nitroimidazoles (e.g., metronidazole) or glycopeptides (e.g., vancomycine), also showed some effect against biofilms of *F. nucleatum* and *P. gingivalis* [393], as well as *S. aureus* [434]. Further, PSs can also be useful to combine with antibiotics in order to restore the susceptibility of bacteria to the latter, notably to last-resort antibiotics. As a recent example, Feng et al. reported the photodynamic inactivation of bacterial carbapenemases, both restoring bacterial susceptibility to carbapenems and enhancing the effectiveness of these antibiotics [396]. However, all combinations of PSs and antibiotics may not be effective, since in some cases antibiotics can have an antagonistic effect on PS activity [435]. Finally, it is noteworthy that some antibiotics can act themselves as PSs; Jiang et al. indeed reported light-excited antibiotics for potentiating bacterial killing via ROS generation [436].

#### 5.2.2. Antifungals

The combination of PSs with antifungals is a powerful approach to thwart growing antifungal resistance, especially to fluconazole, which is commonly observed in *C. albicans* strains [437]. Quiroga et al. showed that sublethal photoinactivation mediated by a tetracationic tentacle porphyrin allowed to reduce the MIC of fluconazole in *C. albicans* [438]. Nystatin, a common antifungal used to prevent and treat candidiasis, was combined with photodithazine-based PS and red light in *C. albicans*-infected mice. The combined therapy reduced the fungal viability and decreased the oral lesions and the inflammatory reaction [54]. Moreover, other antifungals (e.g., terbinafine, itraconazole and voriconazole) combined to ALA reported promising results. These combinations could be alternative methods for the treatment of refractory and complex cases of chromoblastomycosis [401,439].

#### 5.2.3. Other Antimicrobial Compounds

Many other antimicrobial compounds may be used in combination with PSs. Antimicrobial peptides (AMPs) are oligopeptides (commonly consisting of 10–50 amino acids) with high affinity for bacterial cells thanks to an overall positive charge. Their antimicrobial spectrum can be modulated through variation of their amino acids sequence. AMPs encompass a large set of natural compounds that may be used in aPDT. For example, de Freitas et al. reported that sub-lethal doses of PSs (either Ce6 or MB) and aurein peptides (either aurein 1.2 monomer or aurein 1.2 C-terminal dimer) were able to prevent biofilm development by *E. faecalis* [398]. In another study, a membrane-anchoring PS, named TBD-anchor, demonstrated both bacterial membrane-anchoring abilities and ROS production [440]. In a similar way to peptide therapy, LPS-binding proteins were used to improve the contact of PSs with the cytoplasmic membrane of bacteria. For instance, the antibacterial efficacy of a complex consisting of a combination of RB with the lectin concanavalin A (ConA) was demonstrated in a planktonic culture of *E. coli*; ConA-RB conjugates increased membrane damages and enhanced the RB efficacy up to 117-fold [399]. In addition, coupling pump efflux inhibitors, such as CCCP EPI (carbonyl cyanide m-chlorophenylhydrazone), to PSs was also investigated. This highlighted the interest to combine PSs with molecules acting on targets susceptible to induce resistance modulations [40]. Other antimicrobial molecules may be good candidates to be used in aPDT strategies. For example, quinine used in combination with antimicrobial blue light was shown efficient to photo-inactivate MDR *P. aeruginosa* and *A. baumannii* [400]. Some cationic molecules, such as cationic lipids, can have a good affinity for bacterial cell membrane and a good antibacterial activity [126,441]. Thus, PS-amphiphiles conjugates would also deserve to be investigated for aPDT applications.

#### 5.2.4. Viral NPs and Phagotherapy

In recent years, viral NPs (VNPs) deriving from phages, animal, or plant viruses have been proposed as biological vehicles for delivery of PSs. Such carriers can exhibit a series of advantageous properties including natural targeting, easy manufacture and good safety profile [442]. Compared with nanomaterials used as PS carriers, VNPs are natural protein-based NPs that may display higher biocompatibility and tissue specificity. In addition, VNPs can be tuned through genetic and synthetic engineering with appropriate biological and chemical modifications (e.g., surface decoration). Alternatively, the combination of aPDT with so-called phagotherapy may be useful for various reasons, following different strategies. One study suggests that, following a PDT treatment, ROS damages can cause quorum sensing and virulence pathway alterations rendering micro-organisms more susceptible to other therapies. Among these, phagotherapy is known to be modulated by multiple virulence factors [443]. Another approach can consist of conjugating phages with PSs, in order to improve interaction/delivery of the latter into target microorganisms. For example, Dong et al. showed that a phage carrying the chlorophyll-based PS pheophorbide efficiently induced apoptosis in *C. albicans*, thus demonstrating the potential of phototherapeutic nanostructures for fungal inactivation [409]. However, such an approach has to be prudently considered regarding the occurrence of phage resistance already reported in numerous investigations [444].

### 5.3. Combinations of aPDT with Other Light-Based Treatments

Multiple modalities entirely controlled by light stimuli may be combined with aPDT in multifunctional antimicrobial treatments. The following part reports some very recent studies illustrating multiple light-based antimicrobial strategies combined to operate independently, with potential additive or synergistic effects and without reciprocal interferences. This may be obtained with a single PS displaying such multiple properties and/or through combination of PSs with other light-activatable compounds exhibiting complementary properties.

#### 5.3.1. aPDT and Photothermal Therapy

Photothermal therapy (PTT) is a local treatment modality relying on the property of a PS to absorb energy and convert it into heat upon stimulation with an electromagnetic radiation, such as radiofrequency, microwaves, near-infrared irradiation, or visible light. The localized hyperthermia can lead to various damages resulting in microbial inactivation in the treatment area. Following this principle, ruthenium NPs have been used for PDT/PTT dual-modal phototherapeutic killing of pathogenic bacteria [414]. Moreover, GO demonstrated antibacterial effect against *E. coli* and *S. aureus* as a result of both PDT and PTT effects following irradiation with ultra-low doses (65 mW/cm^2^) of 630 nm light [415]. Furthermore, combination of sonodynamic, photodynamic, and photothermal therapies with an external controllable source recently reported against breast cancer [445] may also show promising applications for treating bacterial infection. Mai et al. reported a FDA-approved sinoporphyrin sodium (DVDMS) for photo- and sono-dynamic therapy in cancer cells and photoinactivation of *S. aureus* strains, in in vitro and in vivo models. However, no bacterial sonoinactivation by DVDMS was obtained [446].

#### 5.3.2. aPDT and NO Phototherapy

Combination of aPDT with NO phototherapy is gaining increasing interest for antimicrobial applications [447]. For instance, light-responsive dual NO and ^1^O_2_ releasing materials showed phototoxicities against *E. coli* [355,417]. More recently, Parisi et al. developed a molecular hybrid based on a BODIPY light-harvesting antenna producing simultaneously NO and ^1^O_2_ upon single photon excitation with green light for anticancer applications; according to the authors, this system may also act as an effective PS and NO photodonor antibacterial agent [448].

#### 5.3.3. aPDT and Low Laser Therapy

Photobiomodulation (PBM), also called low-level laser therapy, is a non-destructive process that may alleviate pain and inflammation or promote tissue healing and regeneration. The use of this method coupled to aPDT is a very recent approach. A concomitant use of aPDT and PBM was reported as an adjunct treatment for palatal ulcer [449]. In a clinic-laboratory study, aPDT and PBM showed similar improvement in gingival inflammatory and microbiological parameters compared with conventional treatment [450]. More recently, some benefits of this combined therapy were reported such as the modulation of inflammatory state, pain relief, and acceleration of tissue repair of patients contracting *herpes simplex labialis* virus or orofacial lesions in patients suffering from COVID-19 [451,452,453].

### 5.4. Coupling of aPDT with Other Physical Treatments

#### 5.4.1. aPDT and Sonodynamic Therapy

Sonodynamic therapy (SDT), combining so-called sonosensitizers (SS) and ultrasounds (US) is a relatively new approach for treating microbial infections [454,455]. The ultrasonic waves have the property to induce a cavitation phenomenon thus enhancing the efficacy of combined antimicrobial treatments. The rationale for combining PDT with SDT relies on specific advantages of the latter, notably, a deeper propagation of US into the tissue than light; therefore, PDT/SDT may be used to treat deeper lesions in vivo, alleviating the limitations of light propagation and delivery presented by aPDT [405,456]. An approach combining PDT and SDT, called sonophotodynamic therapy (SPDT), has been reported to improve microbial inactivation compared with individual aPDT or SDT [457]. Because of the complicated system of SPDT, its mechanisms have not been clearly revealed yet. Some studies have demonstrated that sonoporation mechanism induced by US improves the transfer of large molecules into the bacteria by forming transient pores. Moreover, US waves could potentiate the microorganisms dispersion in the medium resulting in (i) a better biodisponibility of therapeutic agent and light diffusion and (ii) a reduction of microbial aggregation and networks, such as biofilm [457,458]. The mention of a new class of PS characterized by the dual ability to be activated by both US and light, for SPDT application, has been questioned. Indeed, Harris et al. recently suggested that this specific PS/SS class could be useful for antimicrobial application – beside previously reported anticancer application – with initial investigation using chlorins as dual PS/SS agent [459]. Since this study by Harris et al., a few dual-activated PS/SS have been described that would warrant further investigations [460,461].

#### 5.4.2. aPDT and Electrochemotherapy

Electrochemotherapy, also called pulsed electromagnetic fields (PEMFs) or electropermeabilization, is a method consisting in applying an electrical field to cells in order to enhance their permeability to therapeutic molecules (often chemical drugs or DNA). Combination of PDT with electrochemotherapy has been used many times to treat cancer diseases [14,462,463]. One study showed that, compared with aPDT used alone, hypericin combined with electrochemotherapy allows to achieve more than 2 to 3 log_10_ CFU reduction in *E. coli* and *S. aureus*, respectively [407]. To our knowledge, no other study combining electrochemotherapy with aPDT was recorded since then. However, combination of aPDT and a cell-permeabilization technique with a controllable toxicity degree may be highly relevant since most PSs can act in extracellular medium without having a specific target. Accordingly, electrochemotherapy may allow boosting aPDT activity by promoting PSs internalization into target microorganisms.

### 5.5. aPDT and Other Antimicrobial-Related Therapies

Immunotherapeutic effects may be obtained as a result of PDT itself or due to other treatments used in combination. For instance, Schiff base complexes with differential immune-stimulatory and immune-modulatory activities were reported efficient to eliminate both Gram(+) and Gram(−) bacteria. Furthermore, upon photoactivation, these complexes blocked the production of the inflammatory cytokine TNFα, thereby allowing to treat at once bacterial infections associated with damaging inflammation [403]. One recent study proposed the first application of antimicrobial photoimmunotherapy (PIT) by developing a PS-antibody complex, selective to the HIV antigen anchored to the infected cell membranes [404]. Such an approach supports the therapeutic applicability of PDT against antimicrobial infections, especially those mediated by intracellular pathogens. In addition, photodynamic therapy using PSs at sub-lethal concentrations may exhibit interesting properties for inflammatory and infectious conditions [464]. It was shown effective to alter immune cell function and alleviate immune-mediated disease, to hasten the process of wound healing, and to enhance antibacterial immunity. PDT thus appears as a promising therapeutic modality in infectious and chronic inflammatory diseases such as inflammatory bowel disease and arthritis.

## 6. Other aPDT Perspectives: New Strategies to Efficiently Target Bacteria

Irrespective of the biomedical applications, achieving a precise targeting is crucial to guarantee both efficiency and specificity. For anticancer PDT, many targeting studies have been done, notably for evaluating PSs covalently attached to molecules having affinity for neoplasia or ligands for receptors expressed on tumors. By this way, PSs may be chosen considering primarily their ability to achieve high PDT effects rather than depending on their intrinsic targeting properties. Following the same rationale, aPDT-based combinatory systems can be developed, benefiting from earlier studies performed in multimodal oncology [465].

### 6.1. Aggregation-Induced Emission (AIE) Luminogens

AIE luminogens exhibit, in the aggregated state, nonradiative decay and show bright fluorescence due to the restriction of intramolecular motions [9]. Recently, their interests for antimicrobial applications have been reported, showing the possibility to simultaneously perform detection and image-guided elimination of bacteria for theranostics applications [466]. In comparison with classical PSs, AIE luminogens in an aggregated state do not exhibit self-quenched fluorescence and ROS production is better. For instance, Gao et al. reported a tetraphenylethylene-based discrete organoplatinum(II) metallacycle electrostatically assembled with a peptide-decorated virus coat protein. This assembly showed strong membrane-intercalating ability, especially in Gram(−) bacteria, and behaved as a potent AIE-PS upon light irradiation [467].

### 6.2. Photochemical Internalization (PCI)

PCI may be used to enhance cell internalization of diverse macromolecules. It consists of PDT-induced disruption of endocytic vesicles and lysosomes improving the release of their payloads into the cytoplasm of target cells. Although most PCI applications relate to cancer treatments, PCI could be extended to treat intracellular infections by delivering antimicrobials into infected cells [468]. For instance, Zhang et al. reported PCI as an antibiotic delivery strategy allowing to enhance cytoplasmic release of Gentamicin, to counter intracellular staphylococcal infection in eukaryotic cells and in zebrafish embryos [469].

### 6.3. Genetically-Encoded PSs

Internalization of PSs inside target microorganisms could be facilitated thanks to their conjugation with adjuvants, as mentioned before. Alternatively, it could be possible to use genetically-encoded ROS-generating proteins (RGPs), also called genetically-encoded PSs. Such an approach represents a powerful way to “completely localize” PSs inside target microorganisms for highly specific antimicrobial phototoxicity. Furthermore, in situ production of RGPs allows to enhance interaction with intracellular targets and better control the biodistribution of PSs, while limiting side-effects for the host tissues and environments [470]. To date, two groups of RGPs have been reported; those that belong to the green fluorescent protein (GFP) family and form their chromophores auto catalytically, and those that use external ubiquitous co-factors (flavins) as chromophores [471]. For example, Endres et al. compared eleven light-oxygen-voltage-based flavin binding fluorescent proteins and showed that most were potent PSs for light-controlled killing of bacteria [472].

### 6.4. pH-Sensitive aPDT

Some studies have reported smart photoactive systems consisting in PSs assembled in nanoconjugates with acid-cleavable linkers. For instance, Staegemann et al. described porphyrins conjugated with acid-labile benzacetal linkers and demonstrated the cleavage of the active PS agents from the polymer carrier in the acidic bacterial environment [316]. In addition, photoacids may be useful to design pH-sensitive aPDT systems. Upon light irradiation, such agents promote the spatial and temporal control of proton-release processes and could provide a way to convert photoenergy into other types of energy [473]. Thus, proof-of-concept was reported for the use of reversible photo-switchable chemicals as antimicrobials inducing MDR bacteria photoinactivation mediated by the acidification of intercellular environment [474]. To our knowledge, no studies have yet reported the potential of photoacids in combination with aPDT systems. However, some pH-sensitive PSs can induce remarkable variations of antimicrobial photoinactivation levels under different environmental pH [475]. These observations suggest the potential of photoacids as PDT potentiators for enhanced antimicrobial applications.

### 6.5. DNA Origami as PS Carriers

The quite recent development of DNA origami based on well-established DNA nanotechnology can serve as an excellent scaffold for the functionalization with different kinds of molecules and could be a powerful tool, as described by Yang at al., to study in a real-time conditions the assemble/disassemble of photo-controllable nanostructures [476]. Oligonucleotides organized as DNA origami could thus be used as PS-carrying nanostructures featuring numerous and dense intercalation sites. In addition, the tightly packed double helices can avoid the degradation by DNA hydrolases in the cellular environment. For instance, Zhuang et al. reported the uptake in tumor cells of a PS-loaded DNA origami nanostructure where it generated free radicals, releasing PSs due to DNA photocleavage, and induced cell apoptosis [477]. To our knowledge, such an approach has not yet been investigated for antimicrobial purposes.

## 7. Discussion

Antimicrobial PDT has the potential to fight against a wide spectrum of infective agents, including those resistant to conventional antimicrobials, under non-clinical and clinical settings. Rather than replacement, aPDT may be a complementary approach to reduce the use of current, especially last resort, antimicrobials. This review aims to give a non-exhaustive overview of the diversity and richness of synthetic, natural, or hybrid single PSs and aPDT nanosystems that were recently reported, with respect to their specific advantages, limitations, and possible evolutions. It is noteworthy that many systems and strategies primarily developed for anticancer PDT have been or could be applied—*per se* or following adaptations—to antimicrobial applications.

Beyond the “chemical space” that can be explored with individual PSs, versatile combinations with other compounds can allow the design of multimodular/multimodal systems. Along this line, the various PSs available may be considered as “basic ingredients”. Apart from offering alternative possibilities for overcoming the most common limitations of PSs (i.e., solubility, delivery, and specificity), reasons for implementing PSs in complex systems can be related to (i) the ability to target several types of pathogens at once (extended antimicrobial spectrum), (ii) synergistic antimicrobial effects, (iii) reduction in the dose of each combined component, (iv) beneficial effects in severe poly-pathogenic infections, and (v) reduction in the risk of resistance emergence.

In addition to the many possible variations concerning PSs, optimizations can also consider other critical parameters in PDT, namely light irradiation and oxygenation. The latter was considered for a long time as an indispensable component. However, control of the oxygen level in the aPDT system is questionable. Indeed, additional oxygen-independent phototoxic mechanisms have been reported, for example with psoralens, which can produce more effective aPDT without oxygen [21]. Furthermore, recent studies suggest that various strategies could be used to reduce or bypass the limitations of oxygen and light supplies (read below). All combined, optimizations targeting not only the PSs, but also light irradiation and oxygen supply, could allow to evolve toward integrative, highly sophisticated, antimicrobial photodynamic therapy.

Light irradiation and its various modalities have been reviewed in depth by different authors [12,478,479,480]. Typically, the irradiation in PDT occurs in the UV (200–400 nm) or in the visible light (400–700 nm) with a power of ≤ 100 mW [479]. However, the low light-penetration depth (around mm) and possible occurrence of tissue photodamage limit the applicability of this spectral range for PDT. This can be circumvented by application of a near-infrared (NIR) irradiation (750–1100 nm), particularly via a two-photon excitation, which is emerging for PDT applications [223,481]. Being a third-order nonlinear optic phenomenon and corresponding to the simultaneous absorption of two photons with half the resonant energy, it allows deeper penetration in biological tissues (around 2 cm), lower scattering losses, and a three-dimensional spatial resolution [482]. Light sources are also constantly improving. Laser is an exceptional source of radiation, capable of producing extremely fine spectral bands, intense, coherent electromagnetic fields ranging from NIR to UV. In comparison, LEDs feature other advantages, notably the possibility to be arranged in many ways, in large quantity, for irradiating wide areas while inducing negligible heating [478,479].

Considering that light-emitted sources can become a brake to any PDT applications, several promising options have been envisaged quite recently. Notably, Blum et al. have identified a series of “self-exiting way” that allows to abolish the need of light to achieve efficient PDT [483]. Among them, chemiluminescence was extensively investigated by using luminol or luciferase energy transfer to induce a chemiexcitation of PS as antibacterial therapeutics [484]. In addition, other methods may be based on Cherenkov radiation, which occurs when an emitted charged particle, such as an electron, moves with high speed through a dielectric medium, such as water. Thus, the polarization of electrons in the medium produces electromagnetic waves in the visible wavelengths that could activate PDT reaction. Another way to induce Cherenkov radiation is to use radioactive isotopes with high beta emissions (e.g., ^18^F, ^64^Cu, or ^68^Ga) as an electronic excitation source [483]. The relevance of such approaches for the design of a self-induced aPDT system remains to be evaluated. In addition, another external source of excitation, such as microwaves, could activate photoactive molecules such as Fe_3_O_4_ when they are complexed with carbon nanotubes. This was recently evaluated by Qiao et al. for treating MRSA-infected osteomyelitis [48]. Furthermore, X-ray as an activated source can also facilitate the activation of the PDT system by transferring energy harvested from X-ray irradiation to the PS used [485]. Kamanli et al. compared pulse and superpulse radiation modes, showing that the latter is more effective to produce ^1^O_2_ and *S. aureus* eradication than the former [486].

Oxygen is also a critical limiting factor that determines PDT efficiency, especially in poorly oxygenated environments. However, it is noteworthy that PDT can be achieved in cancers that typically feature a low oxygenation rate. To alleviate the possible limitations of oxygen supply in PDT systems, several strategies have also been recently considered. One is based on catalase grafting to achieve oxygen self-sufficient NPs in order to convert H_2_O_2_ into available dissolved oxygen in the tumor environment. The abundant ROS in tumors compared with normal tissues provide a coherent substrate for catalase and thus allows an improvement of PDT activity [487]. Some multifunctional nanomaterials, called nanozymes, can also be used in combination with PSs to achieve a catalase-like activity supplying an oxygen source for the PS functioning [488,489]. In addition, it was also reported that noble metal NPs – such as Ru, Pt, and Au NPs – exhibit catalase-like nanozyme activities [490]. The use of catalases or nanozymes may have the crucial role of oxygen helpers in aPDT for treating deep infections.

The possibilities to design combinatorial aPDT strategies seem unlimited. Those presented in this review partly overlap with the description of aPDT systems, showing the complexity of any classification process. Awareness and caution may be raised about some sort of “paradox” or “dilemma”; indeed, while the seemingly boundless collection of chemical options and modular tools to develop nanoscale aPDT therapeutics implicitly defines extensive design flexibility, it staggeringly complicates and bewilders at the same time the optimization process aiming to compare, rationalize, and identify the “best” option for each application. Along this line, data concerning structure-activity relationships are usually missing, with not many studies yet dedicated to this matter [11,42,91,491]. Similar to the CLSI guidelines defined for antibiotics, uniform research methodologies would be useful to assay aPDT systems under well-defined standards and guidelines, considering notably (i) illumination settings, (ii) positive controls [492], (iii) microorganisms and cell lines relevant for a given application, and (iv) assessment of antimicrobial efficacy; this would guarantee better-conducted preclinical and clinical trials of aPDT systems used as mono or combinatory therapy. Furthermore, a public database compiling the efficacy/side effects of various systems would also be useful, facilitating meta-analyses for delineating (quantitative) structure/activity relationships and computational simulations [493].

Finally, in view of translation to clinical practice, a series of precautions and potential limitations must also be considered, especially when dealing with combinatory strategies. Beside possible reciprocal interferences (antagonisms) between combined partners, safety and specificity parameters must also be carefully examined. One main advantage of aPDT is the possibility to control the production of ROS thanks to the use of nontoxic PSs triggered with inducers (specific light and free oxygen) and/or enhancers. The vast majority of PSs are considered safe and dark cytotoxic side effects toward non-target eukaryotic cells are rarely reported. However, in many cases, more studies are needed for examining—beyond potential side-effects—other parameters such as bioavailability, biodispersion, persistence in host cells and body, and elimination pathways. With regard to combinatory strategies, it is noteworthy that studies conducted to date were mostly based on in vitro evaluation or using animal models bearing well-defined sites of infection (Table 2). Thus, much more data in preclinical and clinical settings are required to support the actual potential of such strategies. Moreover, considering that many microbial infections are systemic, the use of modalities such as sonodynamic therapy or electrochemotherapy is at present not realistically feasible. Therefore, in spite of significant progress and real promise, further important work/innovations are needed to effectively broaden the range of infectious conditions that could be treated via aPDT approaches. Lastly, cost effectiveness of multiplexed aPDT therapies over monomodal conventional antimicrobial agents is another crucial point to be considered in view of clinical applications. Potent, but too expensive solutions may fail to be used in practice, especially in under-developed countries where conventional antimicrobials will continue to be used, allowing microorganisms to increase in resistance.

## 8. Concluding Remarks

In conclusion, aPDT is a versatile approach that tends to evolve from quite a simple method to reach much higher degrees of complexity, with several expected advantages, but also possible drawbacks or undesirable effects. Among the latter, any direct/indirect impact on AMR should be more thoroughly considered. It is noteworthy that aPDT clinical trials conducted to date evaluated quite simple PSs. Any increase in the complexity of therapeutic systems would lead to an increase in difficulty before being able to reach clinical applications. The development of effective and safe aPDT treatments requires expertise in many fields of research, including biology (microbiology, cell biology, biochemistry, pharmacology), chemistry, physics (optical physics), and engineering. This is even more the case with combinatory strategies involving different modalities as reviewed in this article. Translation to practical applications also implies strong collaborations with the different sectors of health care and pharmaceutical companies. In these conditions, aPDT and its many therapeutic combinations could become a frontline routine treatment to fight against microorganisms possibly responsible for the next healthcare crises [61].

## Figures and Tables

**Figure 1 pharmaceutics-13-01995-f001:**
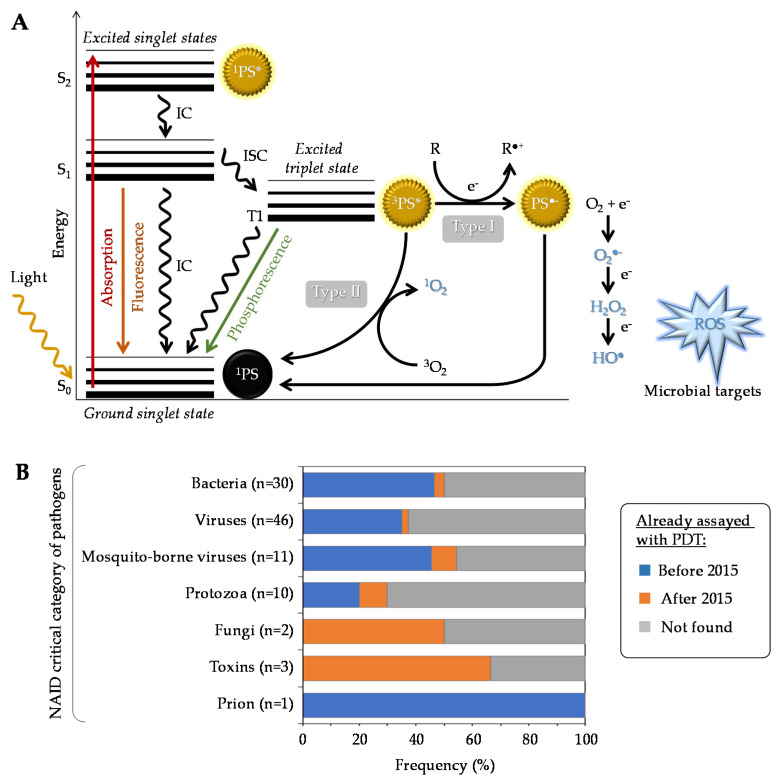
(**A**) Modified Jablonski diagram describing the photochemical and photophysical mechanisms leading to ROS production during PDT. (**B**) Overview of aPDT already applied to the critical category of pathogens, as defined by the NIAID (https://www.niaid.nih.gov/research/emerging-infectious-diseases-pathogens, accessed date: 1 September 2021). For each category, the chart specifies the number and the proportion (percent of pathogens already assayed in at least one aPDT study either before or after 2015).

**Figure 2 pharmaceutics-13-01995-f002:**
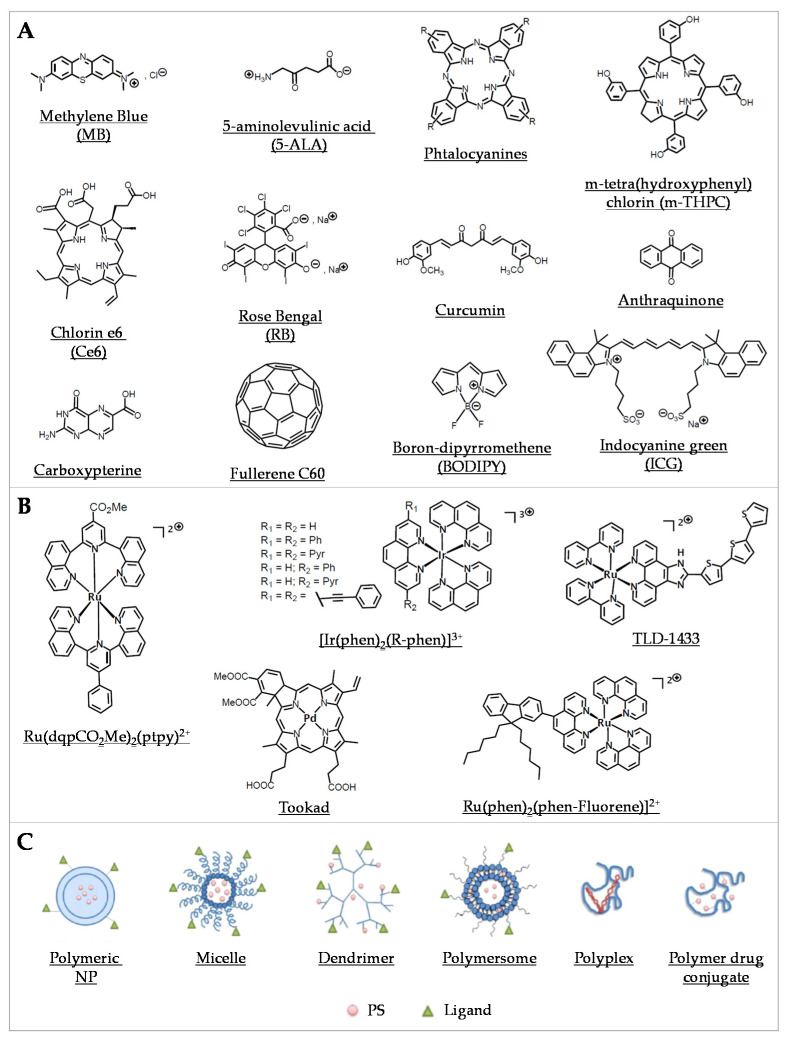
Representative compounds in various classes of PSs used in aPDT. (**A**) Examples of some organic PSs and their derivatives. (**B**) Examples of metallic-based PSs. (**C**) Different types of polymer-based PS carriers, which can be functionalized with ligands for specific target delivery (Adapted from [94], published by MDPI, 2020).

**Table 1 pharmaceutics-13-01995-t001:** List of some recently completed or terminated clinical trials that evaluated aPDT to treat diverse infectious diseases.

Medical Conditions	Target Micro-Organism(s)	Photosensitizer	Trial Phase	Number and Year
**Acne**	*Propionibacterium acnes*	Butenyl ALA	N.A.	NCT02313467, 2014
		Lemuteporfin	Phase 1/2	NCT01490736, 2011
		5-ALA	Phase 2	NCT01689935, 2012
		Methyl aminolevulinate	Phase 2	NCT00673933, 2013
**Dental caries**	*Streptococcus mutans, Streptococcus sobrinus, Lactobacillus casei, Fusobacterium nucleatum, and Atopobium rimae*	TBO	Phase 1	NCT02479958, 2015
		MB	Phase 1	NCT02479958, 2015
	*Aggregatibacter actinomycetemcomitans, Tannerella forsythia and Porphyromonas gingivalis*	N.C.	N.A.	NCT03309748, 2017
**Denture-related stomatitis**	*Candida albicans*	MB	Phase 4	NCT02642900, 2015
**Orthodontic**	N.D.	Curcumin	Phase 1	NCT02337192, 2015
**Peri-implantitis**	N.D.	N.D.	Phase 3	NCT02848482, 2016
**Periodontic**	*Aggregatibacter Actinomycetemcomitans, Porphyromonas gingivalis, Prevotella intermedia, Tannerella forsythia and Treponema denticola*	MB	N.A.	NCT03750162, 2018
		ICG	Phase 2	NCT02043340, 2014
		Methyl aminolevulinate	Phase 2	NCT00933543, 2013
		MB	N.A.	NCT03262077, 2017
		MB	Phase 2	NCT03074136, 2017
		Phenothiazine hydrochloride	Phase 4	NCT03498404, 2018
		TB	Phase 4	NCT03412331, 2018
**Distal subungual onychomycosis**	Fungi infecting nails	5-ALA	Phase 2	NCT02355899, 2015
**Endodontic**	*E. faecalis and C. albicans*	MB	Phase 2	NCT02824601, 2016
**HPV infection**	*Human Papillomavirus (HPV)*	5-ALA	Phase 2	NCT02631863, 2015
**Leg ulcers**	Streptococci, anaerobes, coliform, *S. aureus*, *P. aeruginosa*, yeast, and diphtheroids	PPA904	Phase 2	NCT00825760, 2009

Studies collected from ClinicalTrials.gov (https://clinicaltrials.gov/ct2/results?cond=photodynamic+therapy, Accessed Date: 1 March 2021). ALA, alanine; MB, methylene blue; N.A., not applicable; N.C., not communicated; N.D., not determined; PPA904, 3,7-bis(di-*n*-butylamino)phenothiazin-5-ium bromide; and TB (or TBO), toluidine blue.

**Table 2 pharmaceutics-13-01995-t002:** Examples of recent studies that combined aPDT with other antimicrobial actives or treatments.

**Combination with Antibiotics**	**Target(s)**	**In Vitro and/or In Vivo Effect(s)**	**Reference**
5-ALA + Gentamicin	*S. aureus* and *S. epidermidis*	In vitro: antibiofilm synergistic effect	[392]
Photodithazine + Metronidazole	*F. nucleatum* and *P. gingivalis*	In vitro: improvement of antibiofilm effect	[393]
Ce6 NP + Tinidazole	Periodontal pathogenic bacteria	In vitro: synergistic antiperiodontitis effects; in vivo: reduced adsorption of alveolar bone in a rat model of periodontitis	[394]
MB + Clindamycin/Amoxicillin	*E. coli*	In vitro: enhancement of antibiotic susceptibility following aPDT treatment; in vivo: prolonged survival of infected *G. mellonella* larvae	[43]
MB + Gentamicin	*S. aureus* and *P. aeruginosa*	In vitro: synergistic effect on planctonik cultures of both bacteria; positive effect on *P. aeruginosa* biofilm	[395]
MB + Carbapenem	*S. marcescens*, *K. pneumoniae* and *E. aerogenes*	In vitro: impairment of the enzymatic activity and genetic determinants of carbapenemases; restoration of the susceptibility to Carbapenem	[396]
[Ir(ppy)_2_ (ppdh)]PF_6_) + Cefotaxime	*K. pneumoniae*	In vitro: synergistic aPDI effect with Cefotaxime	[397]
**Combination with other antibacterial compounds**	**Target(s)**	**In vitro and/or in vivo effect(s)**	**Reference**
MB or Ce6 + aurein 1.2 monomer or aurein 1.2 C-terminal dimer	*E. faecalis*	In vitro: prevention of biofilm formation with all treatments; improvement of aurein monomer effect when combined with Ce6-PDT	[398]
RB + Concanavalin A	*E. coli*	In vitro: improvement of RB uptake, increased membrane damages and enhanced PDT effect	[399]
MB@GNPDEX-ConA + Carbonyl cyanide m-chlorophenylhydrazone	*K. pneumoniae*	In vitro: enhancement of the MB-NPs mediated phototoxicity with the efflux pump inhibitor CCCP	[40]
Quinine hydrochloride + antimicrobial blue light	MDR *P. aeruginosa* and *A. baumannii*	In vitro: photo-inactivation of planktonic cells and biofilms; in vivo: potentiation of aBL effect in a mouse skin abrasion infection model	[400]
**Combination with other antifungal treatment compounds**	**Target(s)**	**In vitro and/or in vivo effect(s)**	**Reference**
5-ALA + ITZ, itraconazole; TBF, terbinafine; VOR, voriconazole	*Candida* species, dermatophytes, *A*. *fumigatus* and *F*. *monophora*	In vitro: reduction/improvement of lesions, disappearance of plaque	[401]
Photodithazine + Nystatin	Fluconazole-resistant *C. albicans*	In vitro: reduction of fungal viability, decrease in oral lesions and inflammatory reaction; in vivo: decrease in tongue lesions	[54]
5-ALA + Itraconazole	*Trichosporon asahii*	In vitro: better elimination of planktonic and biofilms fungi than single therapy	[402]
**Combination with immunotherapy**	**Target(s)**	**In vitro and/or in vivo effect(s)**	**Reference**
Schiff base complexes	*E. coli et S. aureus*	In vitro: blockage of the production of inflammatory TNFα cytokine	[403]
Porphyrin + phtalocyanine	HIV-infected cells	In vitro: specific phototoxicity against infected cells	[404]
**Combination with sonodynamic therapy (SDT)**	**Target(s)**	**In vitro and/or in vivo effect(s)**	**Reference**
Ce6 derivative Photodithazine + RB	*C. albicans*	In vitro: inactivation of biofilm (viability and total biomass)	[405]
UCNPs + hematoporphyrin + SiO_2_-RB ^1^	Antibiotic-resistant bacteria	In vitro: greater antibacterial effect with SDT and PDT at once	[406]
**Combination with electrochemotherapy**	**Target(s)**	**In vitro and/or in vivo effect(s)**	**Reference**
Hypericin	*E. coli* and *S. aureus*	In vitro: better bacterial inactivation with combined therapies	[407]
**Combination with viral NPs**	**Target(s)**	**In vitro and/or in vivo effect(s)**	**Reference**
TVP-A (luminogen) + PAP phage	*P. aeruginosa*	In vitro: synergistic bacterial recognizing and killing; in vivo: acceleration of healing rates	[408]
Pheophorbide A (chlorophyll) + JM-phage	*C. albicans*	In vitro: better specificity of PS targeting	[409]
Ru(bpy2)phen-IA + Cowpea chlorotic mottle virus	*S. aureus*	In vitro: targeted bacterial photodynamic inactivation	[410]
**Combination of several PSs**	**Target(s)**	**In vitro and/or in vivo effect(s)**	**Reference**
Carboxypterin + MB	*K. pneumoniae*	In vitro: better biofilm eradication	[411]
Phthalocyanines + Graphene QDs	*S. aureus*	In vitro: better bacterial photoinactivation	[412]
ICG + Metformin + Curcumin	*E. faecalis*	In vitro: better biofilm eradication	[35]
Porphyrin + Phthalocyanine	*Leishmania braziliensis*	In vitro: better assimilation of photo-inactivated parasites by macrophages	[413]
**Combination with photothermal therapy (PTT)**	**Target(s)**	**In vitro and/or in vivo effect(s)**	**Reference**
Ruthenium NPs	Pathogenic bacteria	In vitro: bacterial inhibition; in vivo: reduction of bacterial load and repair of infected wounds	[414]
Graphene oxide	*E. coli* and *S. aureus*	In vitro: efficient vector for both PDT and PTT	[415]
ICG + SPIONs	*E. coli, K. pneumoniae, P. aeruginosa,* and *S. epidermis*	In vitro: antimicrobial and antibiofilm activity at a low dose	[148]
Ag-conjugated graphene QDs	*E. coli* and *S. aureus*	In vitro: efficient photoinactivation by PDT and PTT; in vivo: promoted healing in bacteria-infected rat wounds	[280]
PDPPTT (photothermal agent) + MEH-PPV (PS) ^2^	*E. coli*	In vitro: better inhibition rate than PTT/PDT systems used alone	[324]
Mesoporous polydopamine NPs + ICG	*S. aureus*	In vivo: eradication of *S. aureus* biofilm on titanium implant	[416]
**Combination with NO phototherapy**	**Target(s)**	**In vitro and/or in vivo effect(s)**	**Reference**
N-(3-aminopropyl)-3-(trifluoromethyl)-4-nitrobenzenamine + TMPyP/ZnPc	*E. coli*	In vitro: dual-mode photoantibacterial action	[417]
Sulfonated polystyrene NPs (NO photodonor + porphyrin/phthalocyanine)	*E. coli*	In vitro: strong antibacterial action	[355]
[Ru(bpy)_3_]Cl_2_	*P. aeruginosa*	In vitro: PDT/NO synergistic antibiofilm effect	[418]

ALA, alanine; MB, methylene blue; RB, rose bengal; Ce6, chlorin e6; ICG, indocyanine green; SPION, superparamagnetic iron oxide NP. ^1^, hematoporphyrin monomethyl ether enclosed into yolk-structured up-conversion core and covalently linked RB on SiO2 shell; ^2^, photothermal agent poly(diketopyrrolopyrrole-thienothiophene) (PDPPTT) and the photosensitizer poly(2-methoxy-5-((2-ethylhexyl)oxy)-p-phenylenevinylene) (MEH-PPV) in the presence of poly(styrene-co-maleic anhydride).

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
