# Peer review of "Antimicrobial Photodynamic Therapy: Latest Developments with a Focus on Combinatory Strategies"

_pharmaceutics, 2021, doi:10.3390/pharmaceutics13121995_

Round 1

Reviewer 1 Report

The manuscript “antimicrobial photodynamic therapy: latest developments with a focus on combinatory strategies” is a well written manuscript with appropriate bibliography references and give a wide vision above its uses and undesirable effects.

The authors focused their attention in particular to multicomponent systems. The different types combination are exhaustive presented in order to show both advantages and limits of this strategy.

They, also give an overview of targeting strategy for this technique.

Author Response

We thank Reviewer #1 for his interest in our work and his encouraging comments.

Reviewer 2 Report

This review describes antimicrobial photodynamic therapy focused on the latest developments with including combinatorial strategies. The review is extensive, covering too many topics within the photodynamic inactivation of microorganisms. Therefore, some sections are underdeveloped. However, the manuscript is well organized, although the reader will have to dig deeper into some sections.

The following articles, among others, should be cited and commented:

Section 2.2.

http://dx.doi.org/10.1016/j.jphotobiol.2013.01.005

http://dx.doi.org/10.1016/j.ejmech.2012.10.018

4.2.1.2.

https://doi.org/10.1016/j.molstruc.2021.130012

4.2.2.

https://doi.org/10.1016/j.pdpdt.2018.06.020

4.2.3.1.

https://doi.org/10.1111/php.13465

https://doi.org/10.1039/c9ob02487e

https://doi.org/10.1007/s43630-021-00021-1

4.2.5.3.

10.1021/acsami.0c21723

5.1.2.

https://doi.org/10.1039/c7pp00204a

https://doi.org/10.3390/molecules26195877

5.2.2.

http://dx.doi.org/:10.1016/j.dyepig.2010.04.001

http://dx.doi.org/10.1016/j.pdpdt.2015.10.005

Author Response

Answer: We thank Reviewer #2 for his remarks. As suggested, the manuscript was completed with the following articles:

  • Agazzi, M. L., Almodovar, V. A. S., Gsponer, N. S., Bertolotti, S., Tomé, A. C., & Durantini, E. N. (2020). Diketopyrrolopyrrole–fullerene C60 architectures as highly efficient heavy atom-free photosensitizers: Synthesis, photophysical properties and photodynamic activity. Organic & Biomolecular Chemistry, 18(7), 1449–1461. https://doi.org/10.1039/C9OB02487E
  • Agazzi, M. L., Durantini, J. E., Quiroga, E. D., Alvarez, M. G., & Durantini, E. N. (2021). A novel tricationic fullerene C60 as broad-spectrum antimicrobial photosensitizer: Mechanisms of action and potentiation with potassium iodide. Photochemical & Photobiological Sciences, 20(3), 327–341. https://doi.org/10.1007/s43630-021-00021-1
  • Baigorria, E., Reynoso, E., Alvarez, M. G., Milanesio, M. E., & Durantini, E. N. (2018). Silica nanoparticles embedded with water insoluble phthalocyanines for the photoinactivation of microorganisms. Photodiagnosis and Photodynamic Therapy, 23, 261–269. https://doi.org/10.1016/j.pdpdt.2018.06.020
  • Martínez, S. R., Palacios, Y. B., Heredia, D. A., Aiassa, V., Bartolilla, A., & Durantini, A. M. (2021). Self-Sterilizing 3D-Printed Polylactic Acid Surfaces Coated with a BODIPY Photosensitizer. ACS Applied Materials & Interfaces, 13(10), 11597–11608. https://doi.org/10.1021/acsami.0c21723
  • Palacios, Y. B., Durantini, J. E., Heredia, D. A., Martínez, S. R., González de la Torre, L., & Durantini, A. M. (2021). Tuning the Polarity of Fullerene C60 Derivatives for Enhanced Photodynamic Inactivation. Photochemistry and Photobiology. https://doi.org/10.1111/php.13465
  • Pérez, M. E., Durantini, J. E., Reynoso, E., Alvarez, M. G., Milanesio, M. E., & Durantini, E. N. (2021). Porphyrin–Schiff Base Conjugates Bearing Basic Amino Groups as Antimicrobial Phototherapeutic Agents. Molecules, 26(19), 5877. https://doi.org/10.3390/molecules26195877
  • Quiroga, E. D., Cormick, M. P., Pons, P., Alvarez, M. G., & Durantini, E. N. (2012). Mechanistic aspects of the photodynamic inactivation of Candida albicans induced by cationic porphyrin derivatives. European Journal of Medicinal Chemistry, 58, 332–339. https://doi.org/10.1016/j.ejmech.2012.10.018
  • Quiroga, E. D., Mora, S. J., Alvarez, M. G., & Durantini, E. N. (2016). Photodynamic inactivation of Candida albicans by a tetracationic tentacle porphyrin and its analogue without intrinsic charges in presence of fluconazole. Photodiagnosis and Photodynamic Therapy, 13, 334–340. https://doi.org/10.1016/j.pdpdt.2015.10.005
  • Reynoso, E., Quiroga, E. D., Agazzi, M. L., Ballatore, M. B., Bertolotti, S. G., & Durantini, E. N. (2017). Photodynamic inactivation of microorganisms sensitized by cationic BODIPY derivatives potentiated by potassium iodide. Photochemical & Photobiological Sciences, 16(10), 1524–1536. https://doi.org/10.1039/C7PP00204A
  • Sen, P., & Nyokong, T. (2021). Enhanced Photodynamic inactivation of Staphylococcus Aureus with Schiff base substituted Zinc phthalocyanines through conjugation to silver nanoparticles. Journal of Molecular Structure, 1232, 130012. https://doi.org/10.1016/j.molstruc.2021.130012

Reviewer 3 Report

In the manuscript entitled "Antimicrobial photodynamic therapy: latest developments with a focus on combinatory strategies", the authors reviewed the recent research progress in antimicrobial photodynamic therapy. It provided systematic introduction and professional perspectives for antimicrobial photodynamic therapy, and references for the treatment of growing drug-resistant and multi drug-resistant microorganisms. The manuscript was well organized and convincing. The manuscript can be further processed after minor revision.

It will be better if Type III photochemical pathway could be included in Figure 1.

Typing and grammar should be checked and modified, eg:

Line 53, At the beginning of the XXth century..., should be corrected.

Line 58, While anti-cancer PDT is a clinical reality since 25 years..., did it mean 25 years ago?

Line 105, ...abovementioned..., a space was missing.

Author Response

We also thank Reviewer #3 for his comments and helpful remarks. We agree that it would be satisfactory to implement Figure 1 with the Type III photochemical pathway reported by Hamblin and colleagues. However, the exact mechanism involved is not well-described yet and we did not find any Figure illustrating it. Accordingly, in order to avoid introducing any inaccuracy, we did not modify Figure 1. Beside this, we modified the manuscript considering all the other suggestions received.

Reviewer 4 Report

In this review, the principles of PDT are described. While this information is readily available in numerous review articles, putting it here will provide any ‘new’ investigators with this information so that they do not need to go to original sources. This review provides a summary of progress to date but as an idealized approach to a procedure is clearly unable to deal with systemic infections. 

While some of the data have accrued from strictly in vitro studies, the relevance for the treatment of microbial infections in the clinic is far more complex. For example, the content of section 5.1.2 discusses the ability of sodium azide to promote PDT efficacy. This is a potent poison and would never be used in living animals or people. Potassium iodide and potassium selenocyanate might not be good companions for PDT. Nothing in this section has any implications for clinical PDT.   

Check spelling of ‘Cerenkov’ throughout.  It is also unclear how this effect might be used in antimicrobial protocols which would involve exposing patients to ionizing radiation.

Many of the other procedures will be difficult to translate to clinical practice including sonodynamic effects and electrochemotherapy. Many microbial infections are systemic. How would light be delivered to any sites other than well-defined lesions? All of the trials described in Table 1 involve well-defined sites of infection. So while this approach to microbial infections may have a limited utility, I doubt that it represents anything more. 

Author Response

We thank Reviewer #4 for his critical reading of our manuscript. The correct spelling of “Cherenkov” was checked (we found that it can be spelled in several ways, including “Cerenkov”). We agree that translation of the various strategies reviewed to clinical practice can be questioned at present. However, the review is (i) not dealing with an established field and (ii) it was certainly not mentioned or implied that aPDT can be applied to systemic infections. This we state now explicitly in the abstract, the introduction and the final discussion. We also completely agree that this is a priori not feasible to irradiate affected areas deep in the tissue of patients with UV or visible light. However, light of different wavelengths may in the future afford deeper penetration and possibly may find application in some cases. The status of the investigations conducted (under in vitro and/or in vivo conditions) is clearly stated in corresponding sections for highlighting the development stage in each case. In addition a clarifying note has been added at the end of section 5.1.2. Furthermore, the Discussion was further developed with a dedicated paragraph for examining the “pros and cons” of the strategies reviewed and future research & developments needed before reaching clinical applications.